# RNG105/caprin1, an RNA granule protein for dendritic mRNA localization, is essential for long-term memory formation

Kei Nakayama[1,2,3†], Rie Ohashi[1,2†], Yo Shinoda[4,5], Maya Yamazaki[6], Manabu Abe[6], Akihiro Fujikawa[7], Shuji Shigenobu[2,8], Akira Futatsugi[9], Masaharu Noda[2,7], Katsuhiko Mikoshiba[10], Teiichi Furuichi[4], Kenji Sakimura[6], Nobuyuki Shiina[1,2,3*]

[1]Laboratory of Neuronal Cell Biology, National Institute for Basic Biology, Okazaki, Japan; [2]Department of Basic Biology, SOKENDAI, Okazaki, Japan; [3]Okazaki Institute for Integrative Bioscience, Okazaki, Japan; [4]Department of Applied Biological Science, Tokyo University of Science, Noda, Japan; [5]School of Pharmacy, Tokyo University of Pharmacy and Life Sciences, Hachioji, Japan; [6]Department of Cellular Neurobiology, Brain Research Institute, Niigata University, Niigata, Japan; [7]Division of Molecular Neurobiology, National Institute for Basic Biology, Okazaki, Japan; [8]Functional Genomics Facility, National Institute for Basic Biology, Okazaki, Japan; [9]Department of Basic Medical Science, Kobe City College of Nursing, Hyogo, Japan; [10]Laboratory for Developmental Neurobiology, Brain Science Institute, Wako, Japan

*For correspondence:
nshiina@nibb.ac.jp

†These authors contributed equally to this work

Competing interests: The authors declare that no competing interests exist.

**Abstract** Local regulation of synaptic efficacy is thought to be important for proper networking of neurons and memory formation. Dysregulation of global translation influences long-term memory in mice, but the relevance of the regulation specific for local translation by RNA granules remains elusive. Here, we demonstrate roles of RNG105/caprin1 in long-term memory formation. RNG105 deletion in mice impaired synaptic strength and structural plasticity in hippocampal neurons. Furthermore, RNG105-deficient mice displayed unprecedentedly severe defects in long-term memory formation in spatial and contextual learning tasks. Genome-wide profiling of mRNA distribution in the hippocampus revealed an underlying mechanism: RNG105 deficiency impaired the asymmetric somato-dendritic localization of mRNAs. Particularly, RNG105 deficiency reduced the dendritic localization of mRNAs encoding regulators of AMPAR surface expression, which was consistent with attenuated homeostatic AMPAR scaling in dendrites and reduced synaptic strength. Thus, RNG105 has an essential role, as a key regulator of dendritic mRNA localization, in long-term memory formation.
DOI: https://doi.org/10.7554/eLife.29677.001

## Introduction

The formation of long-term memory, but not short-term memory, requires protein translation in neurons (*Bramham and Wells, 2007*; *Costa-Mattioli et al., 2009*). Gene knockout and administration of drugs for global translational regulators influence, that is, enhance or impair, long-term memory formation (*Costa-Mattioli et al., 2009*). Translation in neurons is regulated not only globally but also locally in dendrites near stimulated postsynaptic sites (*Aakalu et al., 2001*; *Yoon et al., 2016*). This local translation is involved in the regulation of synaptic plasticity and functions, and mediated by dendritic mRNA transport by 'RNA granules', membrane-less macromolecular assemblies of mRNAs, ribosomes, and RNA-binding proteins (*Kiebler and Bassell, 2006*; *Weber and Brangwynne, 2012*;

**eLife digest** Messages pass from one nerve cell to the next across gaps called synapses. The first neuron releases chemical signals from the end of its long, thin nerve fiber. The second receives the message at receptors on branching structures known as dendrites. Each connection has a corresponding bump called a dendritic spine. As animals learn, these can grow larger, strengthening the connection. This is the basis of how memories form.

To strengthen a synapse, the cell must transport the materials to the dendritic spine. The cell makes copies of the genetic instructions to strengthen the synapse in the form of messenger RNA (often shortened to mRNA). But, this happens in the body of the cell, a long way from the dendrites themselves. The mRNA travels from the cell body to the dendrites in collections of molecules referred to as 'RNA granules'.

One of the key components of the RNA granule system is a protein called RNG105/caprin1. Now, Nakayama, Ohashi et al. have engineered mice to delete the gene for RNG105/caprin1, revealing its effect on memory. Mice lacking RNG105/caprin1 struggled to make long-term memories. Unlike their normal counterparts, these mutant mice did not become accustomed to new environments or objects. They also found it more challenging to learn the position of a hidden platform in a water-based maze. Lastly, over time, the mutant mice forgot to be fearful of a dark chamber where they had received a small electric shock.

Memories form in a part of the brain called the hippocampus and the dendritic spines in this region were smaller in mice lacking RNG105/caprin1. Furthermore, when the nerve cells from this part of the brain were grown in Petri dishes, they did not respond normally to stimulation. The dendritic spines of normal cells increased in size, but those on the cells lacking RNG105/caprin1 got smaller compared to normal cells.

A closer look revealed that the distribution of mRNA in brain cells from mice lacking RNG105/caprin1 differed from that of normal mice. Some pieces of genetic information failed to make it from the cell body to the dendrites. This included mRNA involved in making regulators of a component of dendritic spines called the AMPA receptor. The AMPA receptor detects the chemical messenger, glutamate, and is crucial for memory formation.

These findings further our understanding of long-term memory and open the way for future research into human disease. Mutations in RNA granule components, including RNG105/caprin1, have links to conditions such as amyotrophic lateral sclerosis (ALS) and autism spectrum disorder (ASD). Further investigation could reveal new targets for drug treatment.

DOI: https://doi.org/10.7554/eLife.29677.002

*Kedersha et al., 2013*). Some components of RNA granules have been reported to be associated with brain functions in disease, for example, fragile X mental retardation by FMRP deficiency and neurodegenerative diseases such as amyotrophic lateral sclerosis (ALS) and frontotemporal lobar degeneration (FTLD) with aggregation of FUS/TLS and TDP-43 in RNA granules (*Lenzken et al., 2014*; *Santos et al., 2014*; *Ling et al., 2013*). However, little influence on long-term memory formation of knockout in mice for RNA granule components, for example, FMRP, CPEB, Staufen1, Pumilio-2, and G3BP1, has left inconclusive the primary question whether regulation of local translation is required for long-term memory formation (*Consortium TD-BFX, 1994*; *Berger-Sweeney et al., 2006*; *Vessey et al., 2008*; *Siemen et al., 2011*; *Martin et al., 2013*).

RNA granule protein 105 (RNG105, also known as caprin1) is a major RNA-binding protein in RNA granules. RNG105 promotes the assembly of RNA granules and is responsible for the transport of its binding mRNAs in cultured cells (*Shiina et al., 2005*; *Kedersha et al., 2016*; *Shiina et al., 2010*). Knockdown and knockout (KO) of RNG105 in cultured neurons causes a reduction in the synaptic connections on dendrites and the density of neural networks (*Shiina et al., 2010*; *Shiina and Tokunaga, 2010*). A heterozygous nonsense mutation in the *Rng105/caprin1* gene has been found in a human patient with autism spectrum disorder (ASD), and heterozygous KO of *Rng105* gene in mice causes ASD-like behavior (*Ohashi et al., 2016*; *Jiang et al., 2013*). These studies suggested the involvement of RNG105 in higher-order brain functions. However, RNG105 homozygous KO mice are neonatally lethal because of respiratory failure (*Shiina et al., 2010*), which has hampered

the analysis of the physiological impact of severe RNG105 deficiency on learning and memory in adult mice.

Here, we generated RNG105 conditional deletion mice using the Cre/loxP system. The conditional deletion mice (*Camk2a-Cre;Rng105*[f/f]) were viable, and subjected to in vivo analyses of synaptic function, behavioral tests for learning and memory, and genome-wide profiling of somatodendritic localization of mRNAs in the hippocampus. The results demonstrated that RNG105 was an essential element of RNA granules for establishing long-term memory, and suggested RNG105-mediated dendritic localization of mRNAs as an underlying mechanism for AMPA receptor (AMPAR) scaling, synaptic strength and plasticity, and long-term memory formation.

## Results

### Generation of RNG105 conditional deletion mice

To investigate physiological functions of RNG105 in adult mice, we generated RNG105 conditional deletion mice by crossing floxed *Rng105* mice and *Camk2a-Cre* transgenic mice for gene deletion in the central nervous system (*Figure 1A–C*). In *Camk2a-Cre;Rng105*[f/f] mice, exons 5–6 were flanked by loxP sequences and deleted by Cre expression (*Figure 1A*). Because frame shift occurs in the exon 5–6-deleted mRNA, most of the functional domains of RNG105 ranging from the N-terminal coiled-coil domain to the C-terminal RG-rich domain (a.a. 123–707) are deleted (*Shiina et al., 2005*). In addition, most of the exon 5–6-deleted mRNA appeared to be degraded by nonsense-mediated mRNA decay (NMD): in the hippocampus where Cre was highly expressed, expression of *Rng105* transcripts from all exons was reduced to the comparable level to that of exon 5–6 transcripts as judged by RNA-seq analysis (*Figure 1—figure supplement 1A*).

*Camk2a-Cre;Rng105*[f/f] mice were born in the Mendelian ratio and showed no apparent abnormalities just after birth. Although their growth was retarded during the lactation period, their body weight recovered thereafter to 80–90% of that of control (*Rng105*[f/f]) mice (*Figure 1D*). In addition to growth retardation, *Camk2a-Cre;Rng105*[f/f] mice were susceptible to death. However, death was spontaneous, and more than 40% of *Camk2a-Cre;Rng105*[f/f] mice survived for more than 4 months (*Figure 1E*). Thus, RNG105 conditional deletion overcame the neonatal lethality of RNG105 conventional KO.

Western blotting revealed that the expression of RNG105 protein was markedly reduced in the cerebral cortex and hippocampus, but not so much in the cerebellum of *Camk2a-Cre;Rng105*[f/f] mice (*Figure 1F*). Although the anti-RNG105 antibody can recognize truncated RNG105 protein (a.a. 1–122) encoded by the exon 5–6-deleted mRNA, it did not detect any truncated form of RNG105 in the cerebrum of *Camk2a-Cre;Rng105*[f/f] mice (*Figure 1—figure supplement 1B*). This supported the notion that the exon 5–6-deleted mRNAs were degraded by NMD and hardly any truncated RNG105 protein was expressed in *Camk2a-Cre;Rng105*[f/f] mice. Immunostaining of hippocampal slices showed that RNG105 was markedly reduced in the somatic layer (stratum pyramidale [SP]) and dendritic layer (stratum radiatum [SR]) of pyramidal neurons, confirming the reduction of RNG105 expression in neurons of *Camk2a-Cre;Rng105*[f/f] mice (*Figure 1G*).

### RNG105 conditional deletion impairs structural plasticity of dendritic spines

To investigate the impact of RNG105 deficiency on synaptic function, we first examined the morphology of pyramidal neurons and dendritic spines in the hippocampal CA1 region. The density of hippocampal neurons, as judged from nuclear staining, was not affected in *Camk2a-Cre;Rng105*[f/f] mice (*Figures 1G* and *2A*). To trace the morphology of pyramidal neurons, *Camk2a-Cre;Rng105*[f/f] mice were crossed with *Thy1-GFP* transgenic mice. Fluorescence imaging revealed that the length and branching of dendrites were comparable between *Rng105*[f/f] and *Camk2a-Cre;Rng105*[f/f] mice (*Figure 2B−D*). The density of spines on dendrites was also equivalent between the genotypes, but the size of spines was smaller in *Camk2a-Cre;Rng105*[f/f] mice (*Figure 2E−G*). Furthermore, the number of mushroom spines was significantly reduced, which suggested that synaptic strength and/or stimulation-dependent plasticity were attenuated in *Camk2a-Cre;Rng105*[f/f] mice (*Figure 2E,H*).

We then analyzed structural plasticity of dendritic spines using a two-photon glutamate uncaging technique. Hippocampal neurons from floxed RNG105 (*Rng105*[f/f]) mice were cultured in dishes and

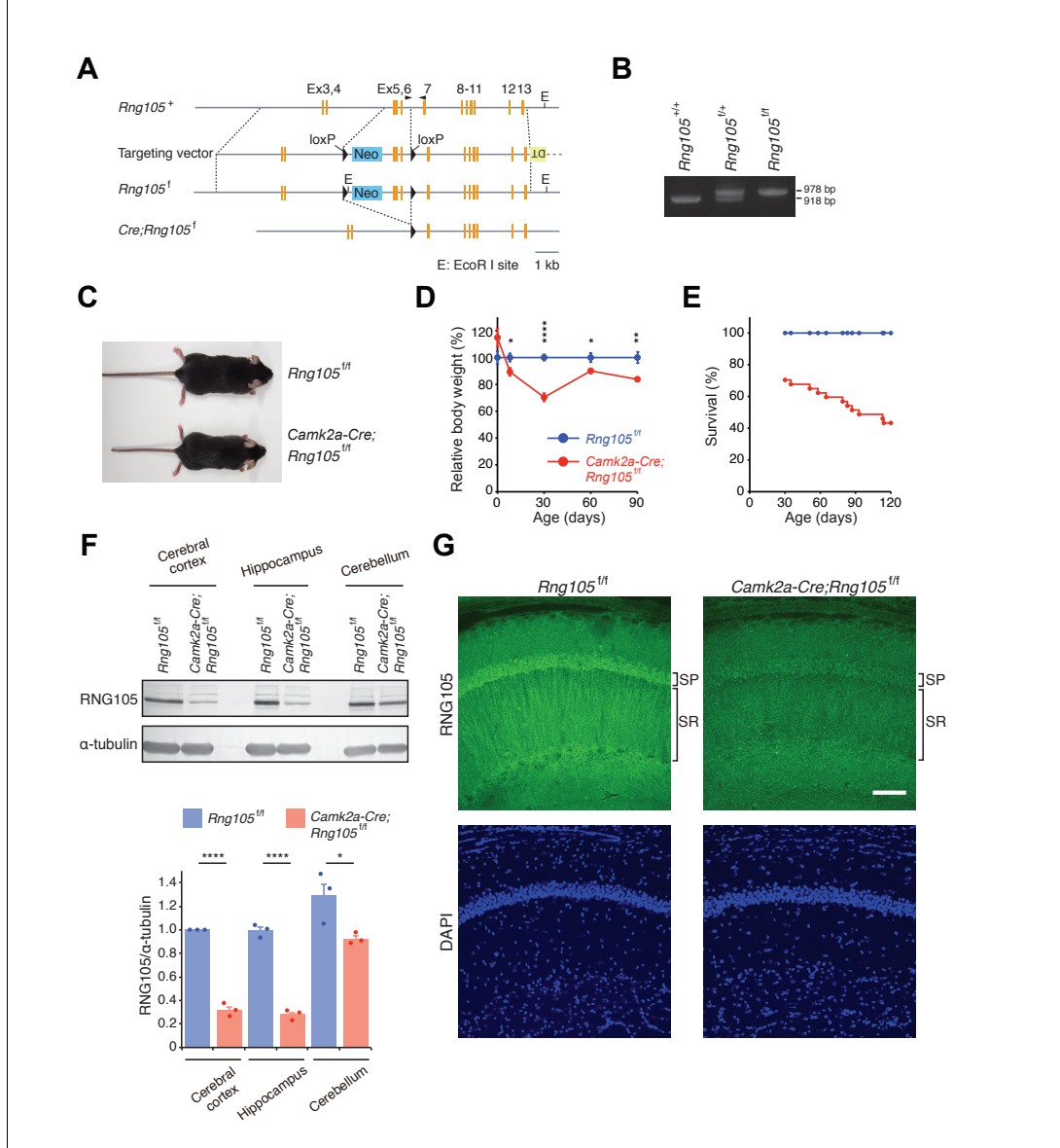

**Figure 1.** Generation of RNG105 conditional deletion mice. (**A**) Gene structure of the *Rng105* genome, targeting vector, floxed *Rng105*, and after excision of the floxed sequence. Arrowheads indicate PCR primers for genotyping. (**B**) PCR analysis of the indicated genotypes. (**C**) Male littermates at postnatal 9 (**P9**) weeks. (**D**) Relative body weight of *Camk2a-Cre;Rng105*^f/f^ mice compared with *Rng105*^f/f^ mice. n = 5 and 3 (P0), 10 and 6 (P8), 27 and 22 (P30), 6 and 6 (P60), 9 and 8 (P90) for *Rng105*^f/f^ and *Camk2a-Cre;Rng105*^f/f^ mice, respectively. *p<0.05, **p<0.01, ****p<0.001 using two-way ANOVA followed by Student's *t*-test. Data are represented as the mean ± s.e.m. (**E**) Survival curves of *Rng105*^f/f^ and *Camk2a-Cre;Rng105*^f/f^ mice. The number of *Camk2a-Cre;Rng105*^f/f^ mice was 70.4% of *Rng105*^f/f^ mice at P30 (n = 402 and n = 283 for *Rng105*^f/f^ and *Camk2a-Cre;Rng105*^f/f^, respectively). Thereafter, survival was analyzed in 13 *Rng105*^f/f^ mice and 26 *Camk2a-Cre;Rng105*^f/f^ mice. (**F**) Western blotting of indicated brain extracts from *Rng105*^f/f^ and *Camk2a-Cre;Rng105*^f/f^ mice for RNG105 and α-tubulin as a control. Bottom, quantification of the RNG105 band intensity normalized to α-tubulin. Data are represented as the mean ± s.e.m. with dot plots of individual values. n = 3. *p<0.05, ****p<0.001 using Student's t-test. (**G**) Staining of hippocampal slices with anti-RNG105 antibody and DAPI. SP, stratum pyramidale; SR, stratum radiatum. Scale bar, 50 μm. See also *Figure 1—figure supplement 1*.

DOI: https://doi.org/10.7554/eLife.29677.003

The following figure supplement is available for figure 1:

**Figure supplement 1.** Reduced expression of RNG105 without the expression of truncated forms in the cerebrum of *Camk2a-Cre;Rng105*^f/f^ mice.
DOI: https://doi.org/10.7554/eLife.29677.004

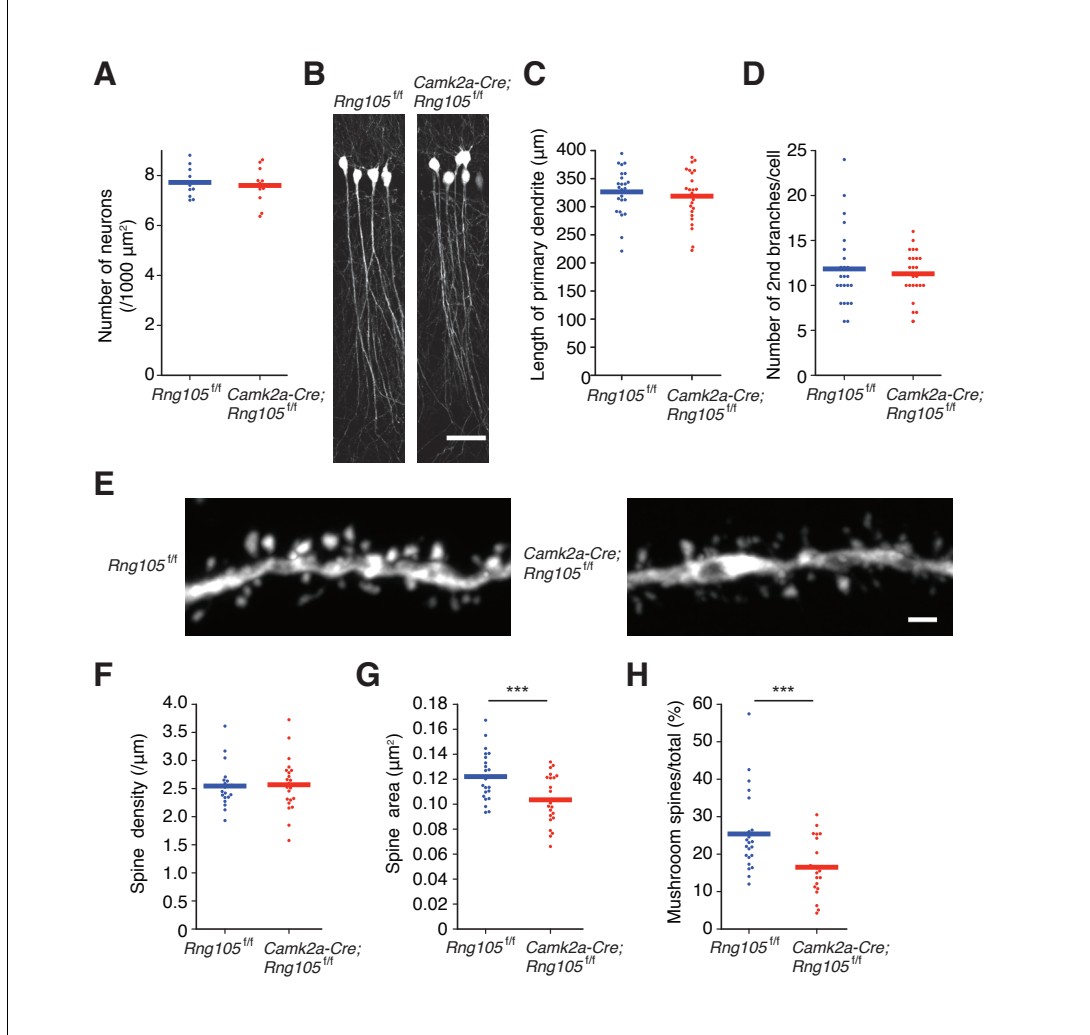

**Figure 2.** RNG105-deficient mice have small dendritic spines. (A) Density of CA1 hippocampal neurons measured by nuclear staining in the SP. n = 11. Student's t-test, p=0.674. (B) CA1 pyramidal neurons imaged with Thy1-GFP. Scale bar, 50 μm. (C and D) Length of primary dendrites (C) and the number of 2nd branches (D) in the CA1 pyramidal neurons. n = 24. Student's t-test, p=0.554 (C), p=0.605 (D). (E) Representative magnified images of a dendrite of a CA1 neuron. Scale bar, 1 μm. (F−H) Spine density (F), spine size (G), and the ratio of the number of mushroom spines to total spines (H) in the CA1 pyramidal neurons. n = 22 dendritic branches. Student's t-test, p=0.857 (F), ***p=0.00380 (G), ***p=0.00278 (H). Data are represented by dot plots with the mean.

DOI: https://doi.org/10.7554/eLife.29677.005

co-transfected with CMV-Cre and mCherry in order to delete the *Rng105* gene and trace the morphology of the neurons, respectively. Immunostaining of the neurons indicated that the expression of RNG105 was significantly reduced in mCherry-positive neurons compared with neighboring mCherry-negative neurons (*Figure 3A,B*). RNG105 deletion did not affect spine density on dendrites, but reduced the size of spines (*Figure 3C,D*), which was similar to the results in the hippocampal slices.

In response to stimulation by glutamate uncaging, single spines close to the uncaging spots were transiently increased in their volume to ~3 fold and sustained an increased state of ~2.5 fold over 60 min in *Rng105*$^{f/f}$ neurons (*Figure 3E,F*). The increase in spine volume during the sustained phase is translation-dependent (*Nishiyama and Yasuda, 2015*; *Tanaka et al., 2008*), which was confirmed by cycloheximide addition (*Figure 3F,G*). In contrast to *Rng105*$^{f/f}$ neurons, *CMV-Cre;Rng105*$^{f/f}$ neurons significantly reduced the spine volume during the sustained phase (*Figure 3E−G*). These results

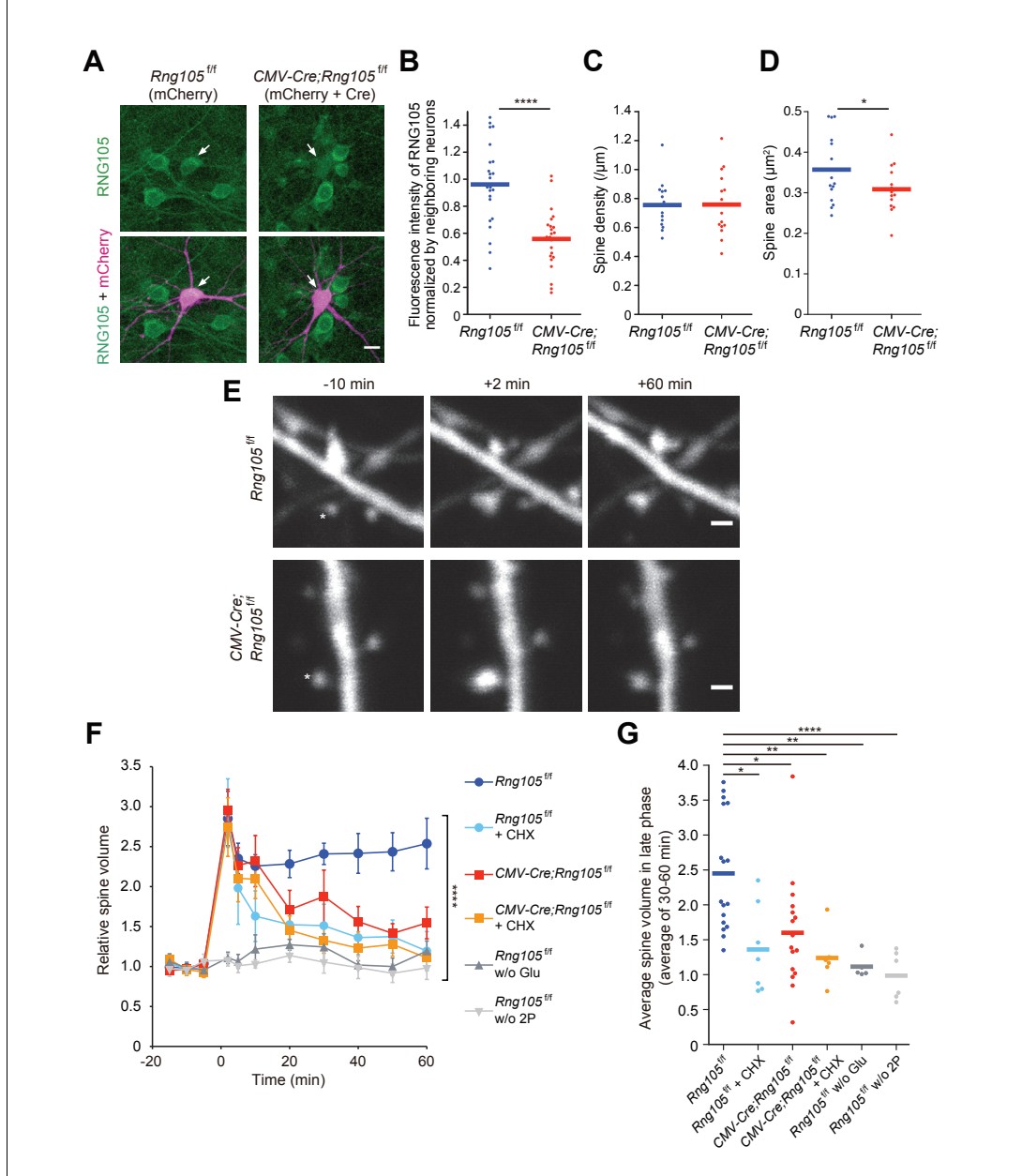

**Figure 3.** RNG105 deficiency impairs structural plasticity of dendritic spines. (A) Fluorescence images of cultured hippocampal neurons from *Rng105*^f/f mice transfected with mCherry (left) or both mCherry and CMV-Cre (right), and immunostained for RNG105. Arrows indicate transfected neurons. Scale bar, 10 μm. (B) Immunofluorescence intensity of RNG105 in the transfected neurons normalized to that in neighboring mCherry-negative neurons. n = 24. Student's t-test, ****p=1.33E-6. (C and D) Density (C) and size (D) of dendritic spines in the transfected neurons. n = 15 neurons. Student's t-test, p=0.483 (C), *p=0.0407 (D). (E) Representative time-lapse images of dendritic spines before and after the induction of structural plasticity by two-photon glutamate uncaging. Asterisks indicate the location of uncaging. Scale bar, 1 μm. (F) Time course of spine volume change in stimulated spines. CHX, addition of cycloheximide; w/o Glu and 2P, mock experiments without caged glutamate or two-photon laser irradiation. n = 17, 7, 16, 6, 5 and 6, from top to bottom, respectively. ****p<0.001 using two-way repeated measures ANOVA. Data are represented as the mean ± s.e.m. (G) Average spine volume over 30–60 min in (F). *p<0.05, **p<0.01, ****p<0.001 using one-way ANOVA followed by Tukey-Kramer test. In B−D and G, data are represented by dot plots with the mean.

DOI: https://doi.org/10.7554/eLife.29677.006

indicated that RNG105 was required for the stimulation-induced structural plasticity of spines in the translation-dependent late-phase LTP.

## RNG105 deficiency reduces synaptic responses to stimulation

To examine whether RNG105 regulates the function of synapses, we measured electrophysiological responses of hippocampal CA1 neurons to stimulation. First, basal synaptic transmission at CA3-CA1 synapses was recorded. The relationship between the magnitude of input stimulation and the post-synaptic responses (amplitude and slope of field excitatory postsynaptic potential [fEPSP]) indicated that RNG105 deficiency reduced both the amplitude and slope of fEPSP to about half of those of control mice (*Figure 4A,B*). Paired-pulse ratio (PPR) was significantly decreased after LTP induction in *Camk2a-Cre;Rng105*$^{f/f}$ mice, but it was not significantly different between the genotypes (*Figure 4C,D*). These results indicated that postsynaptic responses to stimulation were markedly reduced in *Camk2a-Cre;Rng105*$^{f/f}$ mice, suggesting downregulation of AMPARs by RNG105 deficiency.

We also measured LTP at CA3-CA1 synapses. Theta-burst-induced LTP in CA1 neurons revealed that fEPSP was increased from baseline by ~100%, which was comparable between *Rng105*$^{f/f}$ and *Camk2a-Cre;Rng105*$^{f/f}$ mice (*Figure 4E,F*). However, because the absolute amplitude and slope of basal fEPSP in *Camk2a-Cre;Rng105*$^{f/f}$ mice were about half of those in *Rng105*$^{f/f}$ mice, fEPSP after LTP induction in *Camk2a-Cre;Rng105*$^{f/f}$ mice was also about half of that in *Rng105*$^{f/f}$ mice (*Figure 4E,F*). As a result, the absolute amplitude and slope of fEPSP in *Camk2a-Cre;Rng105*$^{f/f}$ mice reached, even after LTP induction, only the same level as the basal fEPSP in *Rng105*$^{f/f}$ mice (*Figure 4E,F*). After the induction of LTP, we often observed epileptic-like repetitive patterns in fEPSP waveforms, which may be a back-propagation of repetitive action potentials, suggesting aberrant neuronal excitation after LTP induction in *Camk2a-Cre;Rng105*$^{f/f}$ mice (*Figure 4C*, inset). Together, in *Camk2a-Cre;Rng105*$^{f/f}$ mice, fEPSP amplitude was reduced both in the steady state and after LTP induction, which may be related to abnormal neuronal excitation and impaired structural plasticity of spines.

## RNG105 conditional deletion mice display impaired long-term memory

Next, we examined whether RNG105 is required for normal behavior in learning and memory tasks. In open-field test, there were no differences in exploratory horizontal locomotion among *Rng105*$^{f/f}$, RNG105 hetero-deletion (*Camk2a-Cre;Rng105*$^{f/+}$) and *Camk2a-Cre;Rng105*$^{f/f}$ mice in the initial trial (*Figure 5A*). The exploratory activity of *Rng105*$^{f/f}$ and *Camk2a-Cre;Rng105*$^{f/+}$ mice was decreased over 3 days, suggesting that the mice were habituated to the novel place with trials. In contrast, *Camk2a-Cre;Rng105*$^{f/f}$ mice did not show such experience-dependent reduction in exploratory activity, suggesting that *Camk2a-Cre;Rng105*$^{f/f}$ mice had problems in becoming habituated to a novel environment (*Figure 5A*).

Habituation is a complex behavior, which is impaired by several factors such as memory deficits, incomplete initial exploration of the entire area because of high level of anxiety, and low level of locomotor activity (*Bolivar, 2009*). Among them, the latter two factors were not likely reasons for the impaired habituation in *Camk2a-Cre;Rng105*$^{f/f}$ mice, because the initial exploratory activity in the open field test was normal and anxiety-like behavior in the light/dark transition test was also normal in *Camk2a-Cre;Rng105*$^{f/f}$ mice (*Figure 5A*; *Figure 5—figure supplement 1A*).

We further conducted a novel object recognition test, which is another test to assess habituation (*Figure 5—figure supplement 1B*). In the first session, mice were habituated to two identical objects and showed no biased preference for either object. In the second session in which one of the objects was replaced by a novel one, *Rng105*$^{f/f}$ mice showed an increased preference for the novel object. In contrast, *Camk2a-Cre;Rng105*$^{f/f}$ mice did not show such an increased preference for the novel object (*Figure 5—figure supplement 1B*). These results supported the notion that habituation, a form of learning and memory, was impaired in *Camk2a-Cre;Rng105*$^{f/f}$ mice.

In the rotarod test, *Rng105*$^{f/f}$ and *Camk2a-Cre;Rng105*$^{f/+}$ mice showed increasing retention time on the rod and decreasing number of falls from the rod over 3 days, indicating that the mice learned the rotarod skill day by day (*Figure 5B*). *Camk2a-Cre;Rng105*$^{f/f}$ mice showed indistinguishable performance from the other genotypes, indicating that RNG105 conditional deletion did not affect motor skill learning. This was consistent with a mild reduction in RNG105 proteins in the cerebellum

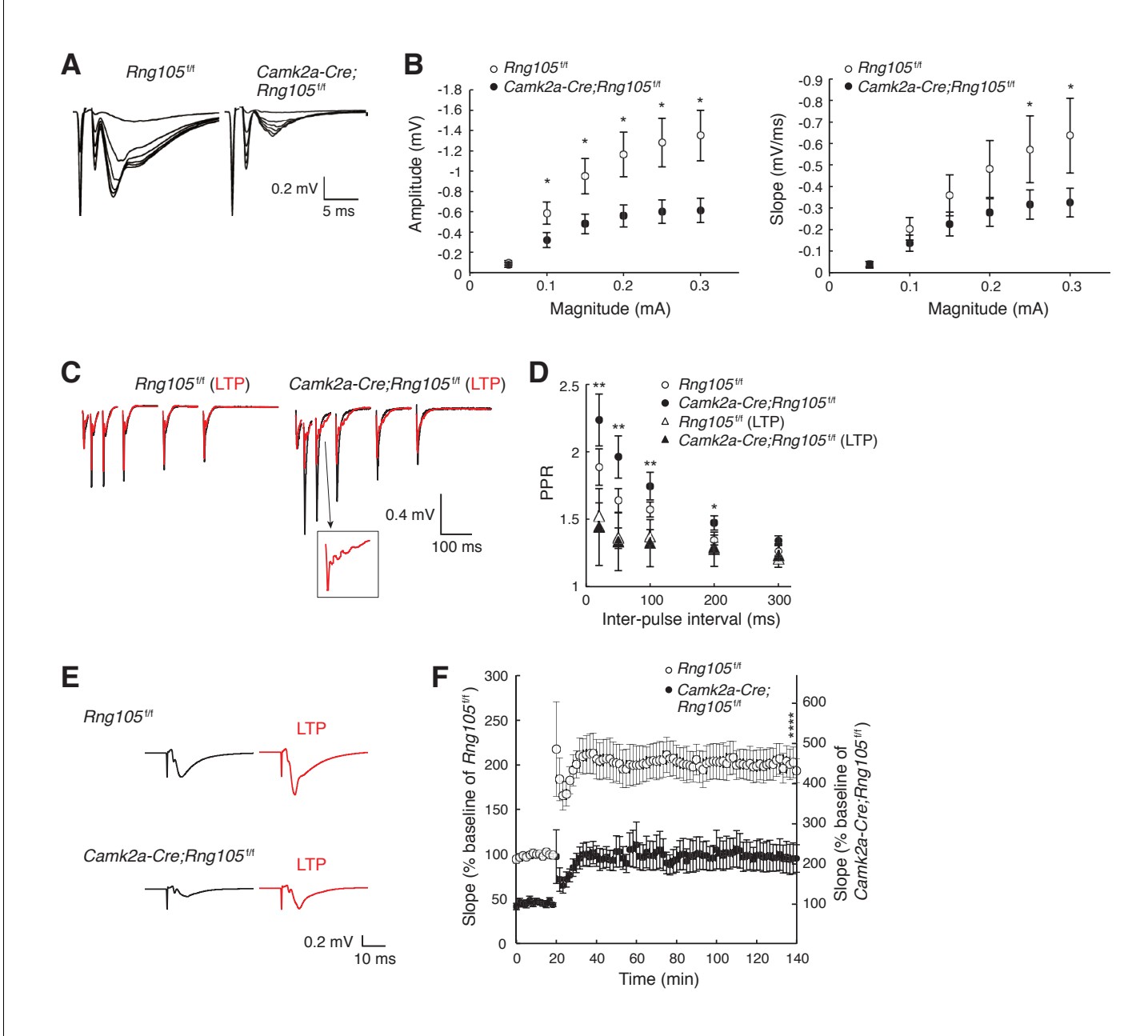

**Figure 4.** RNG105 deficiency reduces fEPSP amplitude in hippocampal CA1. (A) Representative fEPSP traces evoked by different stimulation intensities. (B) Input-output (I/O) relationship between CA3 stimulation magnitude and CA1 fEPSP responses. n = 10 ($Rng105^{f/f}$) and 13 ($Camk2a-Cre;Rng105^{f/f}$). *p<0.05 using two-way repeated measures ANOVA followed by Bonferroni post-hoc t-test. (C) Representative paired-pulse ratio (PPR) with different inter-pulse intervals at CA3-CA1 synapses before (black) and after (red) LTP induction. The inset shows a magnified image. (D) Relationship between inter-pulse intervals and PPR. n = 7 ($Rng105^{f/f}$), 10 ($Camk2a-Cre;Rng105^{f/f}$), 7 ($Rng105^{f/f}$, LTP), and 8 ($Camk2a-Cre;Rng105^{f/f}$, LTP). *p<0.05, **p<0.01, between $Camk2a-Cre;Rng105^{f/f}$ and $Camk2a-Cre;Rng105^{f/f}$ (LTP), using two-way repeated measures ANOVA, Bartlett's test, and one-way ANOVA followed by Tukey-Kramer test. There was no significant difference between the genotypes. (E) Representative fEPSP traces before (black) and after (red) LTP induction. (F) Time course of LTP in CA1 neurons. n = 7. Because the I/O response in $Camk2a-Cre;Rng105^{f/f}$ mice was reduced to 45.1% of that in $Rng105^{f/f}$ mice (at max value), the baseline fEPSP slope for $Camk2a-Cre;Rng105^{f/f}$ mice is set at 45.1% of that for $Rng105^{f/f}$ mice. Left and right scales are for $Rng105^{f/f}$ mice and $Camk2a-Cre;Rng105^{f/f}$ mice, respectively (the right scale is 45.1% of the left scale). ****p=0.0004 using Student's t-test at 140 min with the I/O normalization. Data in B, D and F are represented as the mean ± s.e.m.
DOI: https://doi.org/10.7554/eLife.29677.007

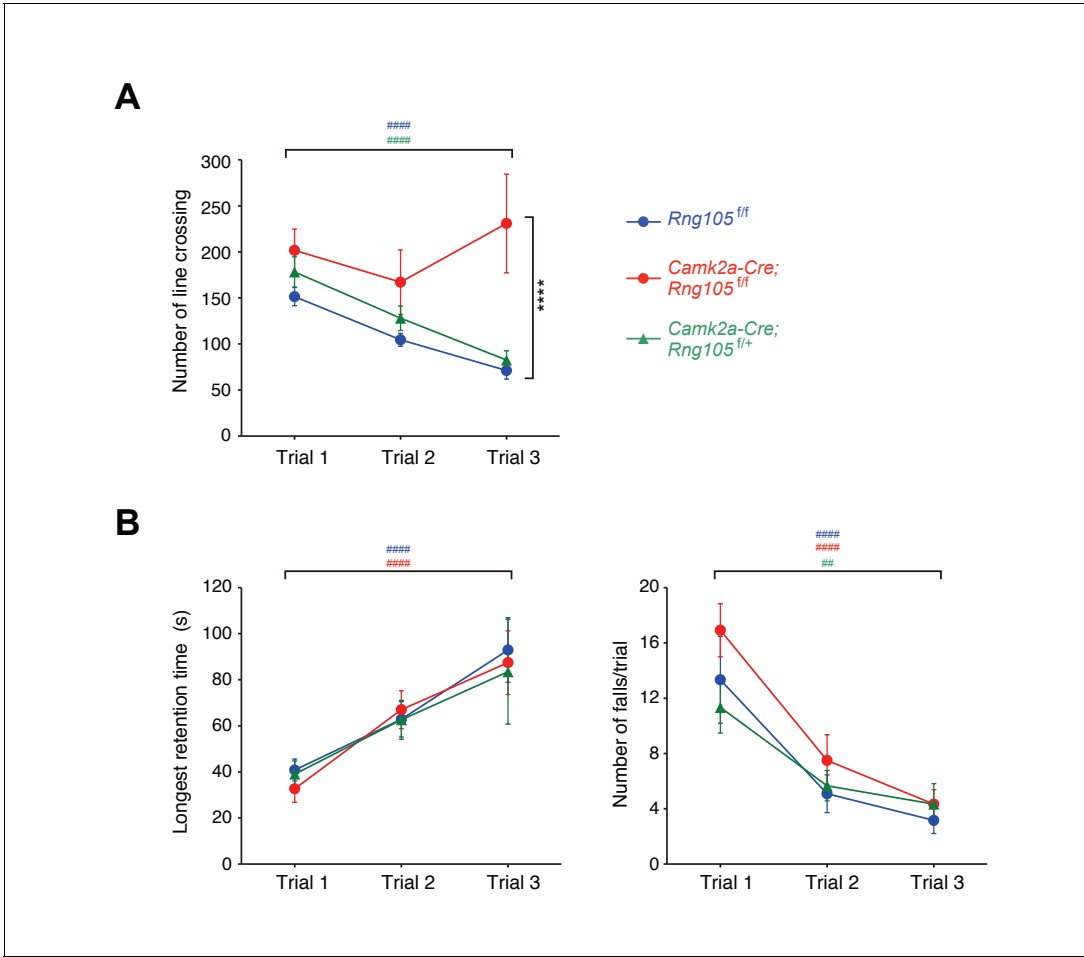

**Figure 5.** Impaired habituation to a novel place and normal motor skill learning in *Camk2a-Cre;Rng105*[f/f] mice. (**A**) Open field test. The number of line crossings during a 5 min trial on 3 consecutive days. n = 14 (*Rng105*[f/f]), 9 (*Camk2a-Cre;Rng105*[f/f]) and 10 (*Camk2a-Cre;Rng105*[f/+]). ****$p < 0.001$ (genotype effect) in the trial 3 using two-way repeated measures ANOVA and one-way ANOVA. (**B**) Rotarod test. Longest retention time on the rod (left) and the number of falls from the rod (right) during a 3 min trial on 3 consecutive days. n = 12 (*Rng105*[f/f]), 12 (*Camk2a-Cre;Rng105*[f/f]) and 6 (*Camk2a-Cre;Rng105*[f/+]). In A and B, ##$p < 0.01$, ####$p < 0.001$ (trial effect) using one-way repeated measures ANOVA for each genotype. '#' is colored corresponding to the genotype. Data are represented as the mean ± s.e.m. See also *Figure 5—figure supplement 1*.

DOI: https://doi.org/10.7554/eLife.29677.008

The following figure supplement is available for figure 5:

**Figure supplement 1.** Normal anxiety-like behavior and reduced novel object preference in *Camk2a-Cre;Rng105*[f/f] mice.

DOI: https://doi.org/10.7554/eLife.29677.009

of *Camk2a-Cre;Rng105*[f/f] mice, as well as a notion that RNG105 may be simply not required for this cerebellum-dependent form of learning and memory.

To test spatial learning and memory, Morris water maze was conducted. Mice were first subjected to a visible platform test. Escape latency of *Camk2a-Cre;Rng105*[f/f] mice was longer than that of *Rng105*[f/f] and *Camk2a-Cre;Rng105*[f/+] mice, but significantly reduced over trials (*Figure 6A*). These results indicated that *Camk2a-Cre;Rng105*[f/f] mice required more training than the other genotypes, but they had escape motivation, vision, and motor skills sufficient to accomplish the task.

The mice were then subjected to a hidden platform test, which necessitates hippocampus-dependent long-term spatial memory (*Figure 6B*). Repeated trials enabled *Rng105*[f/f] and *Camk2a-Cre; Rng105*[f/+] mice to learn the platform location and escape on the platform faster than before the trials. In contrast, the escape latency of *Camk2a-Cre;Rng105*[f/f] mice did not shorten at all over the trials (*Figure 6B*). Following the last trial, a probe test was conducted without the platform. *Rng105*[f/f] and *Camk2a-Cre;Rng105*[f/+] mice intensively searched around the target place (*Figure 6C*). In

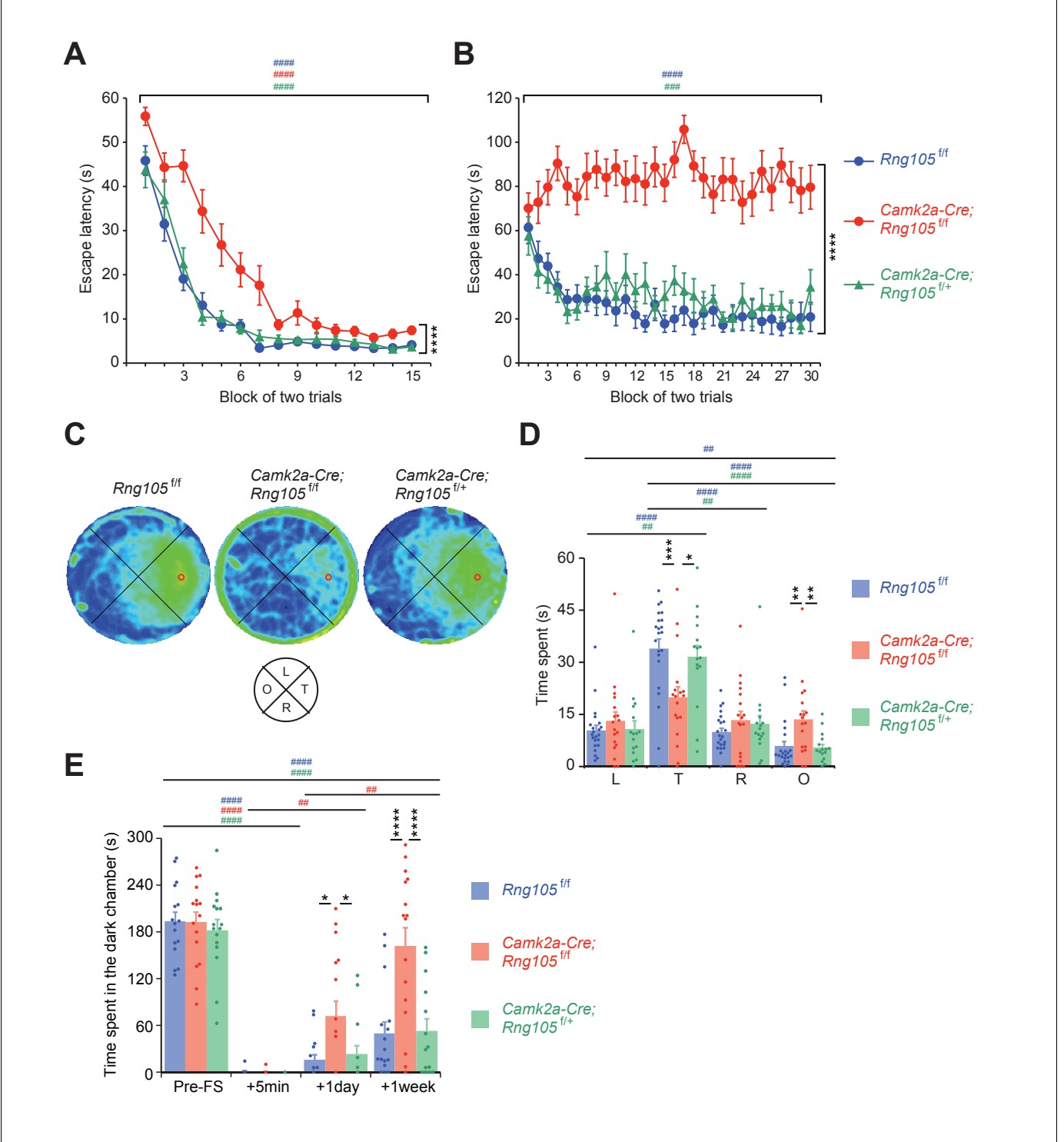

**Figure 6.** RNG105 conditional deletion mice display impaired long-term memory. (A−D) Morris water maze. Escape latency to the platform in visible (A) and hidden (B) platform tasks. Density plot of swim paths averaged from all mice (C) and the time spent in each quadrant (D) during the probe test. In C, occupancy time is indicated by blue (low) to yellow (high) gradient. Red circles in C, platform location; T, target; R, right; L, left; O, opposite quadrants. n = 22 (*Rng105*^f/f), 18 (*Camk2a-Cre;Rng105*^f/f) and 16 (*Camk2a-Cre;Rng105*^f/+). In A and B, ****p<0.001 (genotype effect) using two-way repeated measures ANOVA; ###p<0.005, ####p<0.001 (trial effect) using two-way and one-way repeated measures ANOVA for each genotype. In D, *p<0.05, **p<0.01, ***p<0.005 (genotype effect) using two-way repeated measures ANOVA and one-way ANOVA followed by Tukey-Kramer test; ##p<0.01, ####p<0.001 (quadrant effect) using two-way and one-way repeated measures ANOVA followed by Bonferroni post-hoc paired t-test. (E) Contextual fear conditioning test. Time spent in the dark chamber before foot shock (Pre-FS) and at 5 min, 1 day and 1 week after the conditioning.

*Figure 6 continued on next page*

Figure 6 continued

n = 16 (*Rng105*^f/f^), 16 (*Camk2a-Cre;Rng105*^f/f^) and 14 (*Camk2a-Cre;Rng105*^f/+^). *p<0.05, ****p<0.001 (genotype effect) using two-way repeated measures ANOVA and one-way ANOVA followed by Tukey-Kramer test; ##p<0.01, ####p<0.001 (trial effect) using two-way and one-way repeated measures ANOVA followed by Bonferroni post-hoc paired t-test. '#' is colored corresponding to the genotype. Data are represented as the mean ± s.e.m. In D and E, individual values are plotted by dots. See also *Figure 6—figure supplement 1*.

DOI: https://doi.org/10.7554/eLife.29677.010

The following figure supplement is available for figure 6:

**Figure supplement 1.** *Camk2a-Cre;Rng105*^f/f^ mice exhibit less freezing behavior in a contextual fear conditioning test.

DOI: https://doi.org/10.7554/eLife.29677.011

contrast, *Camk2a-Cre;Rng105*^f/f^ mice showed a circular swimming path along the wall and reduced the search time around the target place (*Figure 6C*). The time that *Rng105*^f/f^ and *Camk2a-Cre; Rng105*^f/+^ mice spent in the target quadrant was significantly longer than in the other quadrants, whereas *Camk2a-Cre;Rng105*^f/f^ mice markedly reduced the time in the target quadrant compared to the other genotypes (*Figure 6D*). These severe phenotypes raised a concern that *Camk2a-Cre; Rng105*^f/f^ mice might be incapable of learning the task. However, if outliers were eliminated (two *Camk2a-Cre;Rng105*^f/f^ mice showing maximum values in L and O quadrants [*Figure 6D*], p<0.01 using Smirnov-Grubbs test), statistical significance was detected in the quadrant effect in *Camk2a-Cre;Rng105*^f/f^ mice (one-way repeated measures ANOVA, F[3,45]=3.234, p=0.0309). Consistently, the density plot of swim path showed weak preference of *Camk2a-Cre;Rng105*^f/f^ mice for the target quadrant (*Figure 6C*), suggesting *Camk2a-Cre;Rng105*^f/f^ mice were able to learn the task. Together, these results indicated that RNG105 was critical for the formation of spatial long-term memory.

Finally, a contextual fear conditioning test (passive avoidance) was conducted. Before receiving foot shock in a dark chamber, all genotypes spent 60–70% of the test time in the dark chamber (*Figure 6E*). At 5 min after the foot shock, none of the genotypes stayed in the dark chamber, indicating that fear-conditioned learning had took place. However, at 1 day after the foot shock, *Camk2a-Cre;Rng105*^f/f^ mice spent a significantly longer time in the dark chamber than the other genotypes. Furthermore, at 1 week after the foot shock, *Camk2a-Cre;Rng105*^f/f^ mice spent as long a time in the dark chamber as before the foot shock (*Figure 6E*). Because *Camk2a-Cre;Rng105*^f/f^ mice spent a comparable time to the other genotypes in the dark chamber before the foot shock, as well as showing normal anxiety-like behavior in the light/dark transition test (*Figure 5—figure supplement 1A*), the increased time in the dark chamber at 1 day and 1 week was considered not to be due to increased anxiety, but reduced long-term memory in *Camk2a-Cre;Rng105*^f/f^ mice. In another contextual fear conditioning test, in which mice received foot shock in a single chamber and their freezing responses were measured in the same chamber at 5 days after the foot shock, *Camk2a-Cre; Rng105*^f/f^ mice showed less freezing behavior than *Rng105*^f/f^ mice (*Figure 6—figure supplement 1*). These results indicated that RNG105 was required for long-term fear conditioning memory formation.

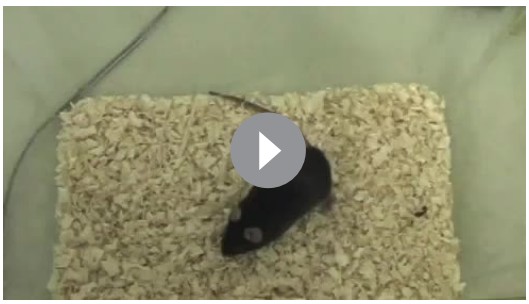

**Video 1.** RNG105 conditional deletion mice exhibit seizures. This movie shows a seizure in a *Camk2a-Cre; Rng105*^f/f^ mouse just after the contextual fear conditioning test (passive avoidance).

DOI: https://doi.org/10.7554/eLife.29677.012

Besides the impairment in memory formation, *Camk2a-Cre;Rng105*^f/f^ mice sometimes exhibited seizures during and just after the Morris water maze and contextual fear conditioning tests (*Video 1*). This phenotype was reminiscent of the epileptic-like fEPSP appeared after relatively intense stimulation of LTP (*Figure 4C*). Taken together, behavioral analyses demonstrated an essential role of RNG105 in the formation of long-term memory.

## RNG105 deficiency reduces asymmetric somato-dendritic localization of mRNAs

To understand the underlying mechanisms by which RNG105 regulates synaptic strength, synaptic structural plasticity and long-term memory

formation, we investigated the impact of RNG105 deficiency on mRNA localization in neurons. Although RNG105 is involved in mRNA transport to dendrites in vitro (*Shiina et al., 2010*), it is unknown whether RNG105 regulates somato-dendritic mRNA localization in vivo, and if so, which mRNAs change the dendritic localization in RNG105-deficient mice.

First, we extracted somatic and dendritic mRNAs from the hippocampal CA1 region in adult mice. The somas of the CA1 pyramidal neurons are aligned in the SP, from which apical dendrites elongate in the SR (*Figures 1G* and *7A*). We microdissected and isolated each layer for the preparation of soma and dendrites of pyramidal neurons (*Figure 7B,C*), as demonstrated in previous studies (*Cajigas et al., 2012*; *Ainsley et al., 2014*). Next, mRNAs extracted from SP and SR of *Rng105*[f/f] mice and *Camk2a-Cre;Rng105*[f/f] mice were subjected to RNA-seq analysis (*Supplementary file 1*). For each mRNA, relative concentration (FPKM: read counts normalized by transcript length) in SR to SP, which we termed the 'dendritic accumulation index (DAI)", was calculated. Statistical analysis identified SP-enriched (low DAI) mRNAs and SR-enriched (high DAI) mRNAs. Because the hippocampus contains not only pyramidal neurons but also other types of cells such as interneurons and glial cells, mRNAs also expressed in these cell types were eliminated from the mRNA lists as in the previous study (*Cajigas et al., 2012*) (*Supplementary file 2A,B*). As a result, we identified 1122 dendritically enriched mRNAs (D-mRNAs) and 2106 somatically enriched mRNAs (S-mRNAs) in control mice (*Figure 7D*; *Supplementary file 3A−C*). The D-mRNAs included already known dendritic mRNAs such as *Camk2a*, *Eef1a1*, *Dlg4*, *Iptr1*, *Arc*, *Shank2*, *Homer2*, and *Limk1* (*Figure 7D*;

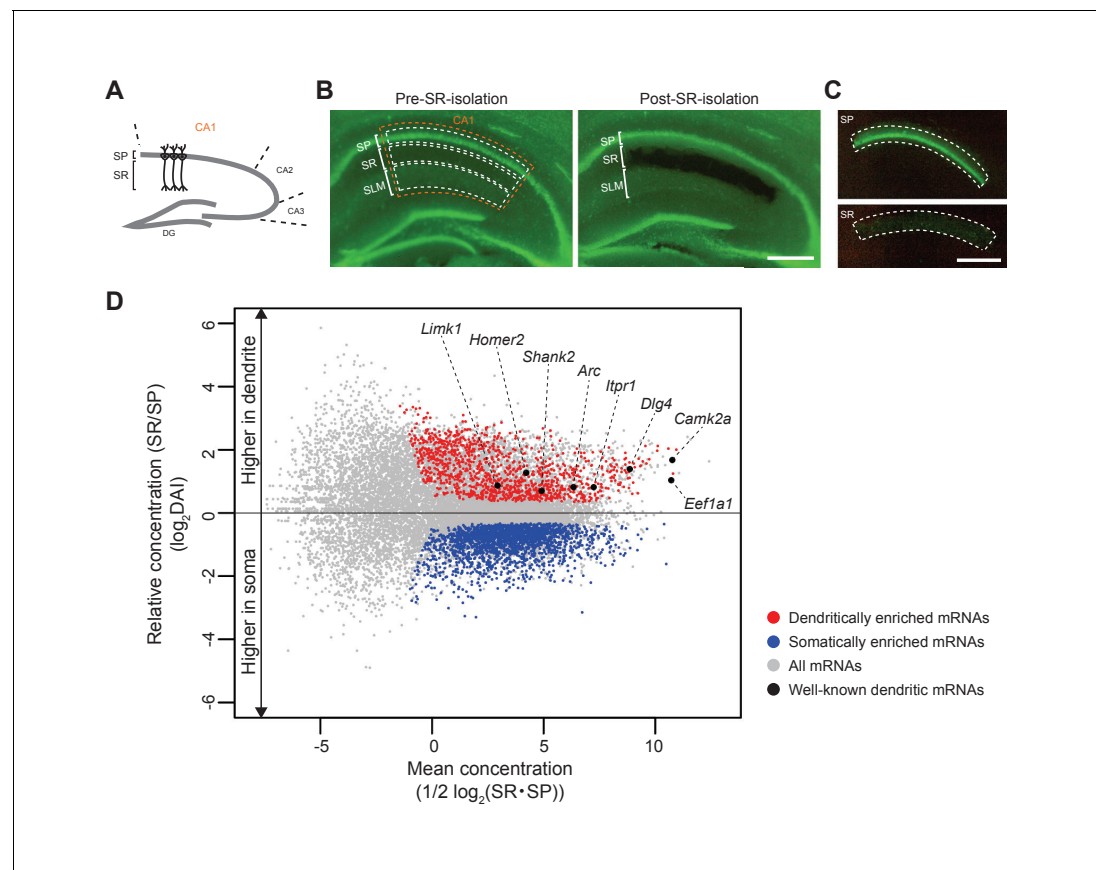

**Figure 7.** Identification of somatically and dendritically enriched mRNAs (S- and D-mRNAs) in the hippocampal CA1. (**A**) Schematic diagram of a mouse hippocampal slice. (**B**) A hippocampal slice from an adult mouse (P12 weeks) before (left) and after (right) isolation of CA1 SR. Nuclei are stained with Yo-Pro1 (green). SP, stratum pyramidale; SR, stratum radiatum; SLM, stratum lacunosum-moleculare. (**C**) Isolated SP (top) and SR (bottom). Scale bars in B and C, 500 μm. (**D**) An MA plot of mRNAs showing their relative enrichment in dendrites (SR) vs. soma (SP) in control (*Rng105*[f/f]) mice. D- and S-mRNAs were identified statistically from three independent experiments with RNA-seq analysis. DAI, dendritic accumulation index.
DOI: https://doi.org/10.7554/eLife.29677.013

*Supplementary file 3A*), suggesting that the strategy was appropriate to detect somato-dendritic mRNA distribution pattern.

Comparison between *Rng105*^f/f and *Camk2a-Cre;Rng105*^f/f mice revealed that the somato-dendritic distribution pattern of mRNAs was different between the genotypes; the variance of DAI of mRNAs was smaller in *Camk2a-Cre;Rng105*^f/f mice than in *Rng105*^f/f mice, and the gap in DAI between D- and S-mRNAs was narrower in *Camk2a-Cre;Rng105*^f/f mice (*Figure 8A*). Variance values ($s^2$) of the DAI of D- and S-mRNAs were 1.945 for *Rng105*^f/f mice and 1.092 for *Camk2a-Cre; Rng105*^f/f mice, which were significantly different ($n = 3,228$, $F_0 = 1.781$, p<0.005). These results indicated an in vivo role of RNG105 to establish the asymmetric localization of mRNAs in the soma and dendrites.

We further analyzed whether changes in the somato-dendritic localization of mRNAs in *Camk2a-Cre;Rng105*^f/f mice were mRNA-selective. For each mRNA, the ratio of DAI in *Camk2a-Cre;Rng105*^f/f mice to *Rng105*^f/f mice was calculated (*Figure 8B*). The ratio was lower than one for majority of, but not all, the D-mRNAs, indicating that RNG105 deficiency reduced the dendritic localization of many, but specific D-mRNAs. To further address mRNA-selectivity, relation between DAI (*Rng105*^f/f) and the ratio of DAI (*Camk2a-Cre;Rng105*^f/f/*Rng105*^f/f) for D- and S-mRNAs was analyzed (*Figure 8C*). D-mRNAs showed large variance and low correlation (|r| = 0.3455). In contrast, S-mRNAs showed smaller variance and high correlation (|r| = 0.7269) (*Figure 8C*). These results indicated that the effect of RNG105 deficiency on dendritic localization of mRNAs varied among D-mRNAs, suggesting RNG105 targets selective D-mRNAs. On the other hand, the relatively uniform effect of RNG105 deficiency on S-mRNAs indicated that the effect was mRNA-non-selective. The apparent increase in DAI of S-mRNAs in *Camk2a-Cre;Rng105*^f/f mice may be mainly a secondary effect of the decrease in DAI of D-mRNAs because the sum of total mRNAs' log(DAI) should be nearly zero.

## Classification of mRNAs whose localization and expression are changed by RNG105 deficiency

To identify the biological categories in which the D-mRNAs are involved, gene ontology (GO) enrichment analysis was conducted. First, D- and S-mRNAs in control mice were classified into GO categories (*Figure 8D*; *Supplementary file 4A−I*). Major categories of D-mRNAs were 'regulation of Arf protein signal transduction', which included GTPase-activating proteins (GAPs) and guanine nucleotide exchange factors (GEFs) of small G protein ADP-ribosylation factor (Arf), and 'structural constituent of ribosomes' which included ribosomal subunit proteins (*Figure 8D*; *Supplementary file 4A, B*). D-mRNAs included in these categories were plotted on MA plots, which indicated that the dendritic accumulation of these mRNAs was comparable to that of well-known dendritic mRNAs (*Figure 9*, cf. *Figure 7D*). Arf is known to regulate membrane trafficking between the cell surface and endosomes, and in particular, Arf6 participates in the surface expression of AMPARs (*D'Souza-Schorey and Chavrier, 2006*; *Jaworski, 2007*; *Oku and Huganir, 2013*; *Zheng et al., 2015*). Arf is also known as a regulator of actin dynamics and dendritic spine formation via Rac1 activation (*D'Souza-Schorey and Chavrier, 2006*; *Jaworski, 2007*). There are various Arf GAPs and GEFs possessing or not possessing a membrane-associated pleckstrin-homology (PH) domain, among which the Arf GAPs and GEFs identified here were mostly the PH domain-possessing types and classified in 'pleckstrin homology' category with other PH domain-containing proteins (*Figure 8D*; *Supplementary file 4E*). The Arf GAPs and GEFs were also classified in 'GTPase regulator activity', which included regulators of other small G proteins such as Ras, Rho, and Rac (*Figure 8D*; *Supplementary file 4G*), known to be involved in actin reorganization and spine morphogenesis (*Nishiyama and Yasuda, 2015*). 'Fc gamma receptor-mediated phagocytosis', 'leukocyte activation', and 'chemotaxis' categories also had high-fold enrichment scores, which contained several overlapping proteins such as PI3 kinase pathway proteins and Rac pathway proteins involved in actin regulation, and extracellular membrane proteins (*Figure 8D*; *Supplementary file 4D,F,H*).

GO enrichment analysis was further conducted on D-mRNAs whose dendritic localization was reduced in *Camk2a-Cre;Rng105*^f/f mice compared with *Rng105*^f/f mice (the ratio of DAI < 0.8). Major categories with high-fold enrichment scores were the same as above, for example, 'regulation of Arf protein signal transduction', 'structural constituent of ribosomes', and 'GTPase regulator activity', suggesting that RNG105 was responsible for the dendritic localization of mRNAs classified in the major categories (*Figures 8E* and *9*; *Supplementary file 4A−I*).

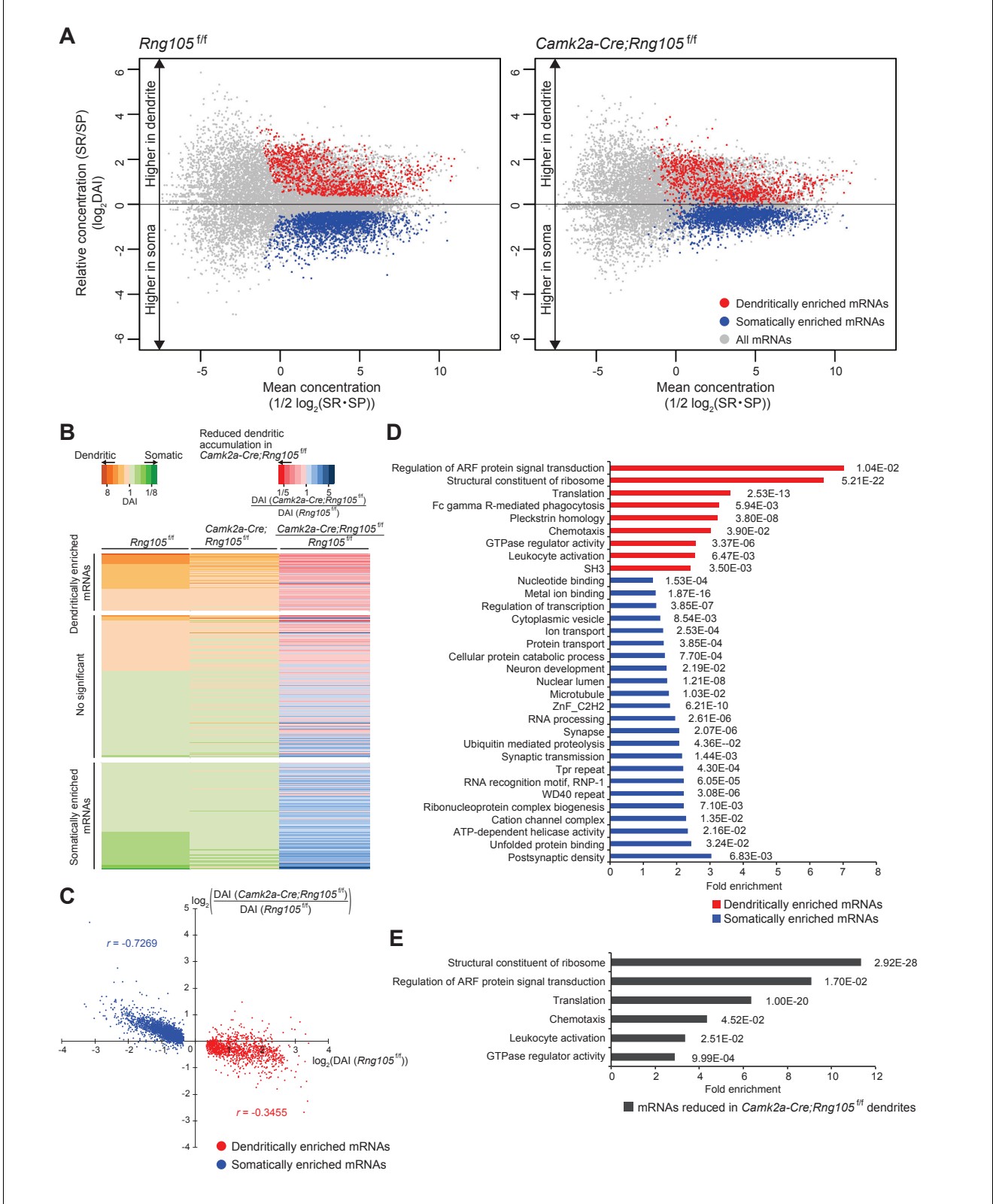

**Figure 8.** Reduced dendritic localization of D-mRNAs in RNG105-deficient mice. (**A**) MA plots of mRNAs in the hippocampal CA1 of *Rng105*^f/f mice (left) and *Camk2a-Cre;Rng105*^f/f mice (right). (**B**) Heat maps showing DAI of each mRNA in *Rng105*^f/f mice (left lane), in *Camk2a-Cre;Rng105*^f/f mice (middle lane), and relative DAI of each mRNA in *Camk2a-Cre;Rng105*^f/f mice compared with *Rng105*^f/f mice (right lane). (**C**) Relationship between somato-dendritic localization and RNG105 conditional deletion-dependent localization changes of D- and S-mRNAs. (**D**) Gene ontology enrichment
*Figure 8 continued on next page*

*Figure 8 continued*

analysis of D-mRNAs (red) and S-mRNAs (blue). (**E**) Gene ontology enrichment analysis of D-mRNAs whose localization to dendrites was reduced in *Camk2a-Cre;Rng105$^{f/f}$* mice (the relative DAI was below 0.8). The numbers in D and E indicate Benjamini values.

DOI: https://doi.org/10.7554/eLife.29677.014

Reduction in the DAI of Arf regulator mRNAs in *Camk2a-Cre;Rng105$^{f/f}$* mice was reminiscent of the results that fEPSP, which is mediated by AMPARs, was reduced in *Camk2a-Cre;Rng105$^{f/f}$* mice (**Figure 4**). We then examined whether the DAIs of mRNAs encoding other regulatory proteins of AMPARs (**Bassani et al., 2013**; **Henley and Wilkinson, 2013**) were also reduced in *Camk2a-Cre; Rng105$^{f/f}$* mice. mRNAs such as *Cnih2*, *Arc*, *Dlg4*, *Pik3r2*, *Camk2a*, and *Cplx2* were identified as D-mRNAs. Furthermore, their DAIs were reduced in *Camk2a-Cre;Rng105$^{f/f}$* mice compared with *Rng105$^{f/f}$* mice (**Supplementary file 5A**). These results indicated that various mRNAs, encoding AMPAR regulators involved in AMPAR surface expression and retention to postsynapses, were localized to dendrites in an RNG105-dependent manner.

By contrast, mRNAs encoding AMPAR subunits themselves (*Gria1-4*) were somatically enriched and their DAIs were not reduced in *Camk2a-Cre;Rng105$^{f/f}$* mice (**Supplementary file 5A**). Notably, the DAI of *Gria2* mRNA was markedly increased, rather than decreased, in *Camk2a-Cre;Rng105$^{f/f}$* mice (**Supplementary file 5A**), suggesting that RNG105 influences specifically, if not directly, the dendritic localization of *Gria2* mRNA.

RNG105 deficiency also influenced, if not directly, the total expression level of some mRNAs, as judged from the S-mRNA concentration (FPKM). mRNAs whose expression was reduced in *Camk2a-Cre;Rng105$^{f/f}$* mice included *Rng105* itself, and notably, a considerable number of immediate early genes (IEGs) such as *Fos*, *Btg2*, *Egr1*, *Egr4*, *Dusp1*, and *Arc* (**Supplementary file 6**). Because the expression of these IEGs was reportedly upregulated by neuronal activation (**Saha et al., 2011**; **Iacono et al., 2013**), these results suggested a reduction in neuronal activity by RNG105 deficiency. In addition, the drastic increase in *Gria2* mRNA localization to dendrites in *Camk2a-Cre;Rng105$^{f/f}$* mice may be also attributed to reduced neuronal activity in RNG105-deficient mice (**Grooms et al., 2006**).

## RNG105 deficiency impairs homeostatic scaling of AMPARs in dendrites

The reduction in fEPSP amplitude and dendritic localization of mRNAs for AMPAR regulators suggested that RNG105 regulates AMPAR scaling. In particular, given the reduction in steady-state fEPSP, we hypothesized that RNG105 may be important for homeostatic scaling of AMPARs. To test this, we analyzed GluR1 and GluR2 cell surface expression in response to activity deprivation with TTX and APV in primary cultured neurons from wild-type (*Rng105$^{+/+}$*) and RNG105 knockout (*Rng105$^{-/-}$*) mice. Surface and total GluR1 and GluR2 were immunostained before and after cell permeabilization, and quantified by counting the number and measuring the fluorescence intensity of their puncta in dendrites (**Figure 10**). After activity deprivation, the number of surface GluR1 puncta normalized to that of total GluR1 was significantly increased, and the intensity of surface GluR1 tended to be increased, in dendrites of *Rng105$^{+/+}$* neurons. By contrast, the activity deprivation-dependent increase in surface GluR1 expression was not observed in *Rng105$^{-/-}$* dendrites (**Figure 10A,C**). Although homeostatic scaling of GluR2 is controversial (**Isaac et al., 2007**; **Wierenga et al., 2005**; **Gainey et al., 2009**), our results indicated that the number of GluR2 surface puncta was increased by activity deprivation in dendrites of *Rng105$^{+/+}$* neurons (**Figure 10B,D**). By contrast, this increase in surface GluR2 puncta was not observed in *Rng105$^{-/-}$* neurons, similarly to GluR1.

Comparison of *Rng105$^{+/+}$* and *Rng105$^{-/-}$* neurons indicated that the number of surface GluR1 in dendrites was significantly reduced in *Rng105$^{-/-}$* neurons (**Figure 10A,C**), confirming the previous results (**Ohashi et al., 2016**). By contrast, GluR2 surface expression in dendrites was not much reduced by RNG105 deficiency (**Figure 10B,D**), which could be due to the increase in *Gria2* mRNA localization to dendrites by RNG105 deficiency (**Supplementary file 5A**) and/or different surface expression pathways of a fraction of GluR2 from that of GluR1 (**Tanaka and Hirano, 2012**).

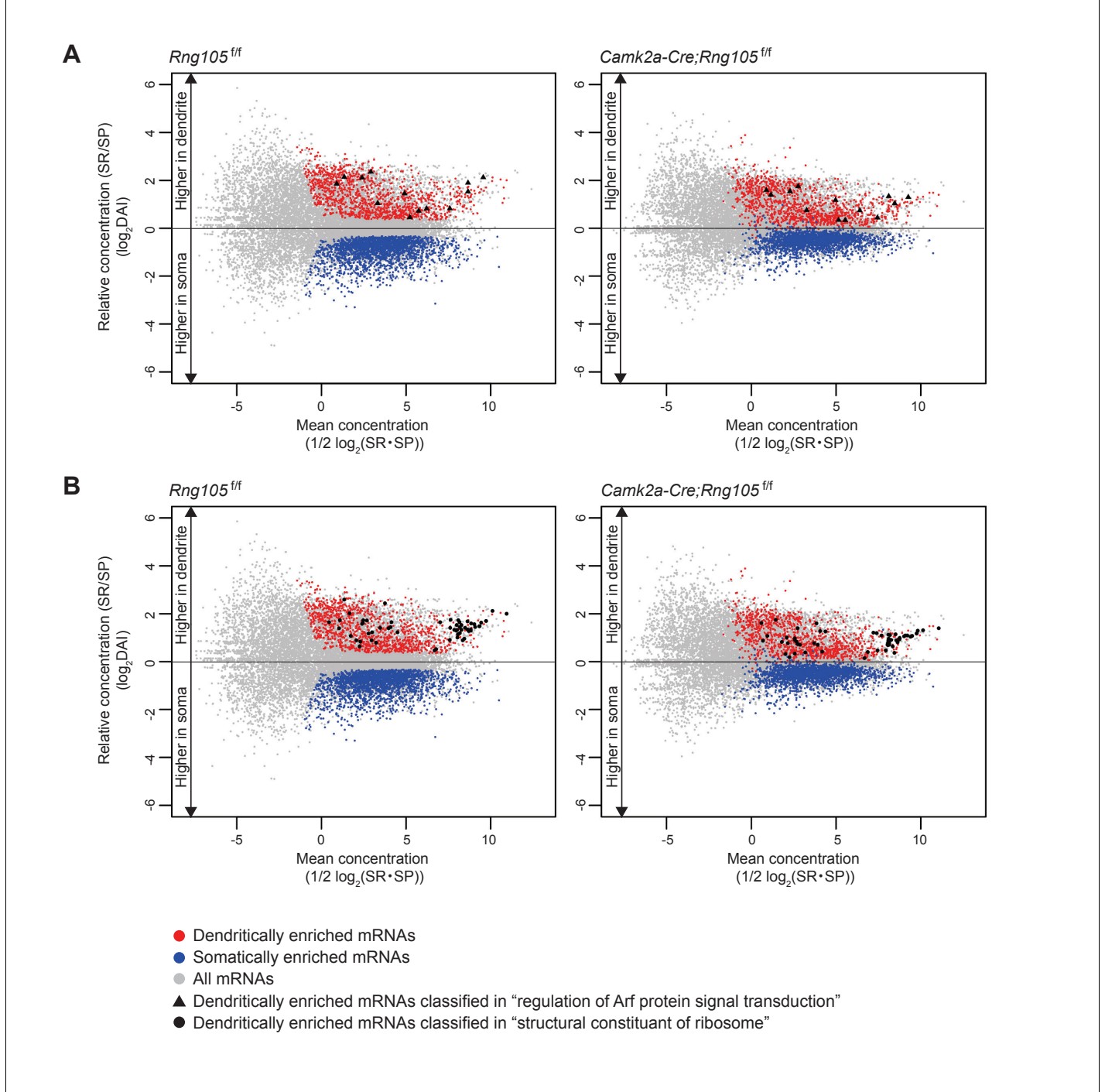

**Figure 9.** Dendritic accumulation of mRNAs classified in 'regulation of Arf protein signal transduction' and 'structural constituent of ribosome' is reduced by RNG105 deficiency. (**A and B**) MA plots of mRNAs classified in 'regulation of Arf protein signal transduction' (**A**) and 'structural constituent of ribosome' (**B**) in *Rng105*[f/f] mice (left) and *Camk2a-Cre;Rng105*[f/f] mice (right). Gray, all mRNAs; red, D-mRNAs; blue, S-mRNAs. D-mRNAs classified in 'regulation of Arf protein signal transduction' (▲) and 'structural constituent of ribosome' (•) are accentuated. DAI, dendritic accumulation index.
DOI: https://doi.org/10.7554/eLife.29677.015

We further conducted biotin labeling of cell surface proteins of cultured neurons followed by immunoblot measurement for GluR1 and GluR2 (*Figure 10—figure supplement 1*). GluR1/2 in total cell lysates was detected as a major band of ~100 kDa, which was reduced in amount after biotin

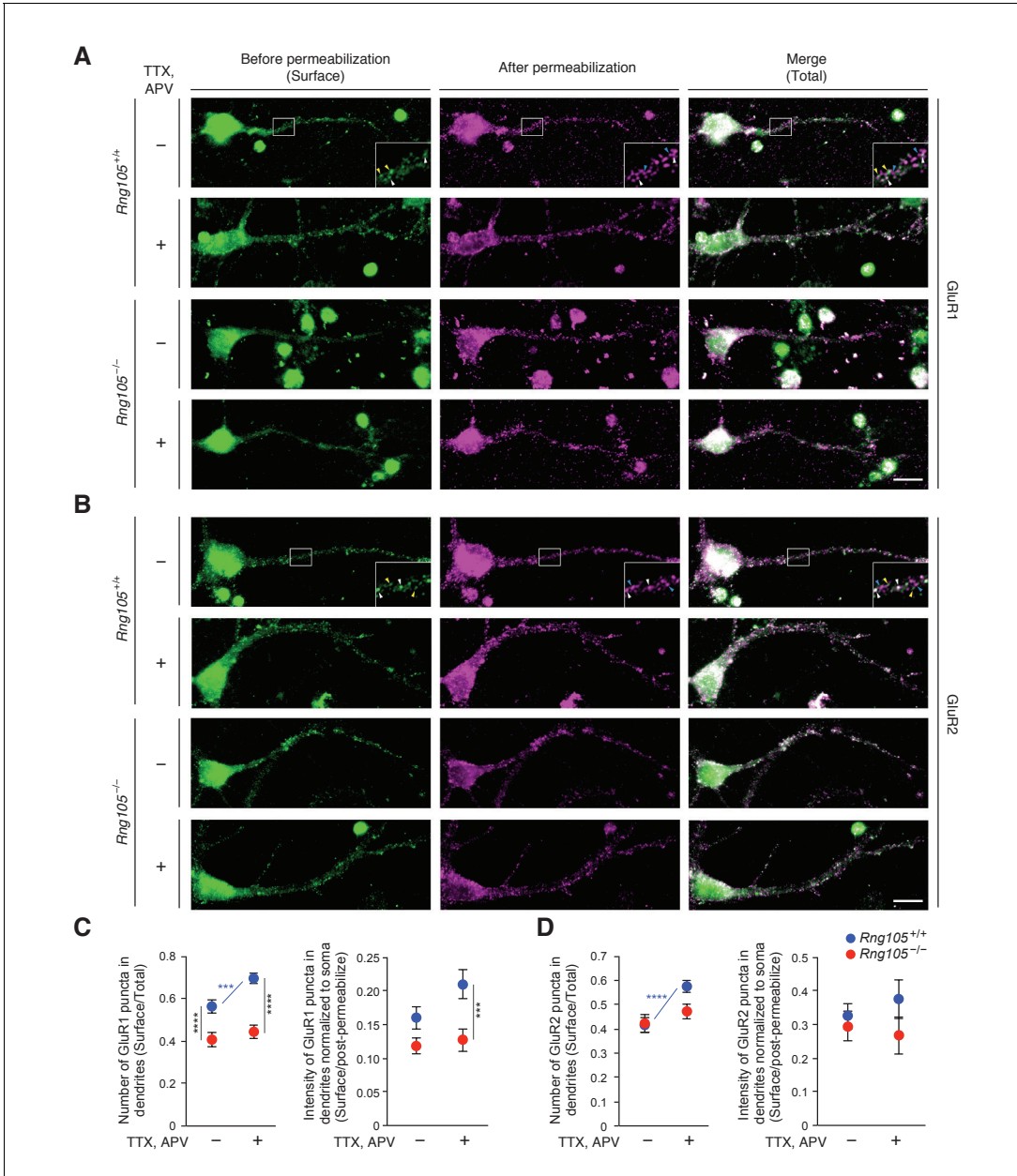

**Figure 10.** RNG105 deficiency impairs AMPAR scaling in response to activity deprivation. (**A and B**) Immunostaining for GluR1 (**A**) and GluR2 (**B**) in cultured neurons (9 DIV) from the cerebral cortex of E17.5 *Rng105*$^{+/+}$ and *Rng105*$^{-/-}$ littermates. The neurons were cultured with (+) or without (-) TTX and APV prior to the staining. GluR1 and GluR2 staining before permeabilization (green, surface proteins), after permeabilization (magenta, intracellular and residual surface proteins), and merged images (total proteins) are shown. GluR1 and GluR2 are distributed in a punctate manner both in the soma and dendrites. The insets show magnified images of boxed areas. Arrowheads denote representative GluR1 and GluR2 puncta which were stained both before and after permeabilization (white), only before permeabilization (yellow) and only after permeabilization (blue). Scale bars, 10 μm. (**C and D**) Quantitative analysis of GluR1 and GluR2 surface expression in dendrites. C, the number of surface GluR1 puncta in dendrites normalized by the number of total GluR1 puncta (left), and fluorescence intensity of surface GluR1 puncta in dendrites normalized by GluR1 fluorescence intensity after permeabilization and in the soma (right). D, the same quantification for GluR2. Data are represented as the mean ± s.e.m. In C, n = 31 (*Rng105*$^{+/+}$, −), 35 (*Rng105*$^{+/+}$, +), 34 (*Rng105*$^{-/-}$, −), and 33 (*Rng105*$^{-/-}$, +) neurons from 4 experiments. In D, n = 39 (*Rng105*$^{+/+}$, −), 40 (*Rng105*$^{+/+}$, +), 39 (*Rng105*$^{-/-}$, −), and 38 (*Rng105*$^{-/-}$, +) neurons from 4 experiments. ***p<0.005, ****p<0.001 using two-way ANOVA followed by post-hoc Student's t-test. See also *Figure 10—figure supplement 1*.

DOI: https://doi.org/10.7554/eLife.29677.016

The following figure supplement is available for figure 10:

**Figure supplement 1.** RNG105 deficiency impairs AMPAR scaling in response to activity deprivation (biotinylation assay).

*Figure 10 continued on next page*

*Figure 10 continued*

DOI: https://doi.org/10.7554/eLife.29677.017

labeling, accompanied by an increase in the amount of an upper band (*Figure 10—figure supplement 1A*). The upper band, but not the lower band, bound to avidin agarose beads, indicating that the upper band was biotin-labeled and mobility-shifted surface GluR1/2, whereas the lower band was non-labeled intracellular GluR1/2 (*Figure 10—figure supplement 1A−C*). Because the upper band was also detected in control lysates without biotinylation, the band was considered to contain a non-specific protein(s) as well as biotinylated GluR1/2.

The most obvious difference between TTX/APV-treated and untreated neurons was the less amount of the lower GluR1 band in TTX/APV-treated neurons from $Rng105^{+/+}$ mice, but not from $Rng105^{-/-}$ mice, which suggested that TTX/APV treatment reduced intracellular GluR1 in $Rng105^{+/+}$ neurons, but not in $Rng105^{-/-}$ neurons (*Figure 10—figure supplement 1B*). Then we quantified the intensity of the lower band in the cell lysate and the upper band in the avidin agarose-bound fraction, and calculated the ratio of surface to intracellular GluR1 and GluR2 (*Figure 10—figure supplement 1D,E*). The ratio of surface/intracellular GluR1 was increased in TTX/APV-treated neurons compared to untreated neurons from $Rng105^{+/+}$ mice. Compared to $Rng105^{+/+}$ neurons, the ratio of surface/intracellular GluR1 was lower and was not significantly increased by TTX/APV treatment in $Rng105^{-/-}$ neurons (*Figure 10—figure supplement 1D*). We noted that the GluR1 band intensity was lower both in the total lysates and avidin-bound fractions in $Rng105^{-/-}$ neurons than in $Rng105^{+/+}$ neurons (*Figure 10—figure supplement 1B*, upper panel), which may be because of the reduced density of neural networks in $Rng105^{-/-}$ primary cultured neurons (*Shiina et al., 2010*). This weaker band intensity in $Rng105^{-/-}$ neurons made it difficult to identify differences in the surface/intracellular ratio between the genotypes at a glance, but when twice the volume of samples from $Rng105^{-/-}$ neurons was loaded on gels, the difference in the surface/intracellular ratio between the genotypes was obvious: the intensity of the lower GluR1 bands was higher, whereas that of the upper GluR1 bands was lower, in $Rng105^{-/-}$ neurons than in $Rng105^{+/+}$ neurons (*Figure 10—figure supplement 1B*, bottom panel). As for GluR2, although statistical significance was not detected, the surface/intracellular ratio of GluR2 tended to be increased by TTX/APV treatment (*Figure 10—figure supplement 1E*). These results were consistent with the results of the immunofluorescence imaging of GluR1 and GluR2. Taken together, these results indicated that RNG105 impacts on homeostatic scaling of AMPARs in dendrites, which is coupled to basal synaptic function and also influences synaptic strength in activated state.

## Discussion

Translation in neurons is required for long-term memory formation, but the relevance of the regulation of mRNA transport and local translation to long-term memory formation has remained unclear. This study demonstrated essential roles of an RNA granule protein RNG105 in the regulation of synaptic strength, structural plasticity of spines, and long-term memory formation. The loss of synaptic strength and long-term memory in *Camk2a-Cre;Rng105*$^{f/f}$ mice was likely attributed to reduced dendritic localization of mRNAs, which included mRNAs for various regulators of AMPAR surface expression and retention to postsynapses. Consistently, AMPAR homeostatic scaling in dendrites was impaired by RNG105 deficiency, which was related to the decreased setting level of fEPSP amplitude in the steady state. The decreased setting level of fEPSP may reduce excitatory transmission in neurons even after LTP induction, because of unchanged (not upregulated) LTP efficiency in RNG105-deficient mice, which will impair long-term memory formation. Together, this study revealed physiological roles of RNG105 in mice, and suggested cellular mechanisms linking RNA granule functions with long-term memory formation.

Studies on learning and memory in KO mice of other RNA granule components, for example, FMRP, Pumilio2, CPEB1, Staufen1, G3BP1, and GLD-2 have been reported. However, these KO mice did not show apparent influence on long-term memory formation in the Morris water maze or contextual fear conditioning (*Consortium TD-BFX, 1994*; *Berger-Sweeney et al., 2006*; *Vessey et al., 2008*; *Siemen et al., 2011*; *Martin et al., 2013*; *Mansur et al., 2016*). Although KO mice of Ataxin-2, another RNA granule component, showed impaired contextual fear conditioning,

their long-term memory in the Morris water maze was normal (*Huynh et al., 2009*). These studies have left unanswered whether RNA granules are involved in long-term memory formation. Compared with these KO mice, RNG105 conditional deletion mice displayed remarkable impairment of long-term memory formation. The different phenotypes between RNG105 conditional deletion mice and the other KO mice of RNA granule components may be attributed to the existence and non-existence of alternative factors. Pumilio1 and 2, Staufen1 and 2, and G3BP1 and 2 are paralogs, and the paralog proteins have redundant functions and target an overlapping set of mRNA (*Kedde et al., 2010*; *Galgano et al., 2008*; *Park and Maquat, 2013*; *Furic et al., 2008*; *Matsuki et al., 2013*). CPEB1 has three paralogs, CPEB2−4, which are not functionally redundant with CPEB1 because they do not bind to the cytoplasmic polyadenylation element (CPE) sequence. However, mRNA polyadenylation by CPEB1 may be redundant with other polyadenylation mechanisms (*Villalba et al., 2011*). FMRP (FMR1) has two paralogs, FXR1 and FXR2. FMR1 and FXR2 function cooperatively and the phenotype of single KO of each gene in mice was less severe than that of double KO mice (*Zhang et al., 2009*). These studies suggest that KO of single genes could be compensated by alternative factors and/or not completely impair the cooperative activity of the factors involved.

Another explanation for the different phenotypes between RNG105 conditional deletion mice and the other KO mice may be that the RNA granule components other than RNG105 are not essential for long-term memory formation. However, this is not unlikely because, in *Drosophila*, mutants of FMRP, Staufen, Pumilio, CPEB1 (Orb), GLD-2 (Wispy), Ataxin-2 (Atx2), and DDX6 (Me31B) have been reported to show defects in long-term memory (*Bolduc et al., 2008*; *Dubnau et al., 2003*; *Pai et al., 2013*; *Kwak et al., 2008*; *Sudhakaran et al., 2014*). The long-term memory deficits in *Drosophila* may be because alternative paralogs of these RNA granule components do not exist in *Drosophila*. In addition, because family proteins are less in *Drosophila* than in mice in general, proteins encoded by target mRNAs of RNA granule components could have less alternative family proteins in *Drosophila* than in mice. Thus, mutations in single genes could lead to severer defects in long-term memory in *Drosophila*.

In contrast to the RNA granule components described above, RNG105 may not have alternatives. Although RNG105 (caprin1) has a paralog, RNG140 (caprin2), in mice, RNG105 and RNG140 are localized to different kinds of RNA granules, and knockdown phenotypes of RNG105 and RNG140 in cultured neurons were not compensated by each other (*Shiina and Tokunaga, 2010*). In addition, because RNG105 targets (affects the localization of) more than a thousand mRNAs, the probability may be large that mRNAs encoding long-term memory-related factors and their alternative factors are simultaneously affected by RNG105 deficiency. Thus, RNG105 deficiency may have large impact on the function of RNA granules and therefore the formation of long-term memory.

This study applied for the first time the technique of the somato-dendritic mRNA identification with RNA-seq to mutant animals. The RNA-seq revealed that the loss of RNG105 reduced the asymmetric somato-dendritic localization of mRNAs in vivo. Furthermore, the RNA-seq identified multiple mRNAs whose dendritic localization was reduced by RNG105 deficiency, which included mRNAs encoding regulators of the cell surface expression and postsynaptic retention of AMPARs, for example, Arf regulator mRNAs. In addition, the identified mRNAs included those encoding $Na^+/K^+$ ATPase subunit isoforms and $K^+$ channel subunits (*Supplementary file 5B*), which was consistent with the previous in vitro study (*Shiina et al., 2010*). Proteins encoded by these mRNAs are involved in the control of membrane potential, and thereby may be associated with fEPSP amplitude and epileptic-like EPSP. Furthermore, the identified mRNAs included those encoding regulators of Ras, Rho, and the PI3 kinase and Rac pathway proteins, involved in actin reorganization and spine formation (*Nishiyama and Yasuda, 2015*; *Sala and Segal, 2014*), which may be associated with the impaired structural plasticity of spines in RNG105-deficient mice. mRNAs for ribosomal subunit proteins were the major dendritic mRNAs and also reported in the previous studies (*Cajigas et al., 2012*; *Ainsley et al., 2014*). However, whether locally translated ribosomal proteins are involved in ribosome biogenesis or in other biological processes, which could be associated with RNG105-deficient phenotypes, remains elusive. Thus, RNG105-dependent dendritic mRNAs included various mRNAs whose encoded proteins are involved in AMPAR localization, membrane potential control, and actin reorganization. Even if the reduction of each mRNA in dendrites could have a small influence on synaptic functions, integration of the reduction of these mRNAs could have a large impact on it.

RNG105 deficiency reduced homeostatic scaling of AMPARs and steady-state fEPSP, and also spine structural plasticity and fEPSP amplitude after LTP induction. These lines of evidence indicate that RNG105 is critical for synaptic functions both in the homeostatic phase and during LTP, which is consistent with the involvement of translational regulation in both the processes (*Nishiyama and Yasuda, 2015*; *Tanaka et al., 2008*; *Cajigas et al., 2010*). Although fEPSP amplitude after LTP induction was small, LTP appeared intact in *Camk2a-Cre;Rng105*^f/f mice. This could be because translation may not be inhibited although the material (mRNA) is reduced in the dendrites of *Camk2a-Cre;Rng105*^f/f mice, which enables the delivery of locally synthesized proteins, even if at a low level, to synapses in the late phase of LTP. Another, though not mutually exclusive, explanation is that translation deficiency may not necessarily cause the decline of LTP in the late phase, but limit the absolute amplitude of EPSP to a certain level in the late phase of LTP. The mechanism of the EPSP limitation is likely coupled with spine size reduction because spine size is tightly correlated with AMPAR expression level in the spine (*Kopec et al., 2007*; *Matsuzaki et al., 2001*). If EPSP amplitude is above the limit in the early-phase LTP, EPSP will decline in the late phase, whereas if EPSP amplitude is already low in the early phase, EPSP will be retained in the late phase. In *Camk2a-Cre;Rng105*^f/f mice, translation deficiency and spine size reduction in dendrites could limit the absolute amplitude of EPSP. However, because the basal EPSP amplitude of *Camk2a-Cre;Rng105*^f/f mice is low, EPSP amplitude after LTP induction could not reach the limit level even if LTP occurs normally. As a result, LTP may appear to be sustained in the late phase, but because of the low absolute amplitude of EPSP, long-term memory may be affected. A similar phenotype to *Camk2a-Cre; Rng105*^f/f mice, that is, long-term memory deficits with low steady-state EPSP and intact LTP, has been reported in other mice such as aged mice, prion-infected mice, chronic stressed mice, and Rett syndrome model (MeCP2-null) mice (*Burke and Barnes, 2006*; *Mallucci et al., 2007*; *Kallarackal et al., 2013*; *Dani and Nelson, 2009*). In these mice, similarly to *Camk2a-Cre;Rng105*^f/f mice, impaired long-term memory formation may be attributed to reduced excitatory transmission in neurons.

There are increasing number of evidence that RNA granules are associated with mental disorder and neurodegenerative diseases. However, the primary question whether RNA granules are required for the formation of long-term memory has been unclear. This study demonstrated that an element of RNA granules, RNG105/caprin1, was required for long-term memory formation, and dendritic localization of mRNAs as an underlying mechanism for AMPAR-dependent synaptic strength and long-term-memory formation.

## Materials and methods

### Ethics statement

All animal care, experiments and behavioral testing procedures were approved by the Institutional Animal Care and Use Committee of the National Institutes of Natural Sciences, and performed in accordance with the guidelines from the National Institutes of Natural Sciences, Niigata University and the Science Council of Japan.

### Generation of RNG105 conditional deletion mice

To generate a *loxP*-flanked (floxed) *Rng105* construct, we isolated the *Rng105/caprin1* gene by PCR from a genomic DNA library of C57BL/6 mice. A DNA fragment, which carried a 34 bp *loxP* sequence and a neomycin resistance gene (*Neo*) flanked by two *frt* sites, was inserted into the site 227 bp upstream of exon 5. The other loxP site was introduced into the site 282 bp downstream of exon 6. The targeting vector contained exons 5 and 6 of the *Rng105* gene flanked by *loxP* sequences, 7.33 kb upstream and 5.8 kb downstream homologous genomic DNA fragments, and the diphtheria toxin (*DT*) gene for negative selection (*Figure 1A*). The targeting vector was introduced into C57BL/6 ES cells (RENKA), and homologous recombinants were selected by G418 resistance and identified by Southern blotting analysis of genomic DNA after EcoRI digestion. ES cell clones with the correct recombination were injected into eight-cell stage embryos of CD-1 mice to generate chimeric mice, which were mated with C57BL/6 mice to obtain heterozygous offspring (*Rng105*^f/+). Homozygous floxed *Rng105* mice (*Rng105*^f/f) were obtained by mating heterozygotes. Floxed *Rng105* mice were further crossed with *Camk2a-Cre* transgenic mice (C57BL/6-TgN[a-CaMKII-nlCre]/

20, RIKEN RBRC00254) to obtain heterozygous conditional deletion mice (*Camk2a-Cre;Rng105*$^{f/+}$). Homozygous conditional deletion mice (*Camk2a-Cre;Rng105*$^{f/f}$) and control mice (*Rng105*$^{f/f}$) were obtained by crossing *Camk2a-Cre;Rng105*$^{f/+}$ female and *Rng105*$^{f/f}$ male mice because Cre is expressed in male germ cells as well as in the central nervous system under the control of the *Camk2a* promoter. Genotyping was performed by PCR with primers 5'-AGATGGCTTTTCTTC TGCCA-3' and 5'-CTGGAAAACACGCTCAACAA-3', which amplified a 918 bp product from the wild-type *Rng105* allele and a 978 bp product from the floxed *Rng105* allele; and primers 5'-GTCGA TGCAACGAGTGATGA-3' and 5'-AGCATTGCTGTCACTTGGTC-3', which amplified a 291 bp product from the *Cre* transgene.

The *Thy1-GFP* transgenic mice (Tg[Thy1-EGFP]MJrs/J) were purchased from Jackson Laboratory (Bar Harbor, ME, USA). We crossed *Camk2a-Cre;Rng105*$^{f/+}$ female and *Thy1-GFP;Rng105*$^{f/f}$ male mice to obtain *Thy1-GFP;Rng105*$^{f/f}$ control mice and *Thy1-GFP;Camk2a-Cre;Rng105*$^{f/f}$ RNG105 conditional deletion mice.

## Cell culture

In the glutamate uncaging experiments, dissociated hippocampal neurons were prepared from *Rng105*$^{f/f}$ embryos at embryonic day 17–18 (E17–18). Neurons were plated at a density of $1.6 \times 10^6$ cells/cm$^2$ onto poly-D-lysine-coated coverslips in glass-bottomed dishes (MatTek, Ashland, MA, USA) in Neurobasal-A medium (Thermo Fisher Scientific, Waltham, MA, USA) containing B-27 supplement (Thermo Fisher Scientific), 0.5 mM glutamine and 25% Neuron culture medium (Wako Pure Chemical Industries, Osaka, Japan). Cultures were incubated at 37°C in a 5% CO$_2$ incubator. The neurons were transfected with plasmids at 6 days in vitro (DIV) using conventional calcium-phosphate transfection method.

In the GluR1 and GluR2 immunostaining and biotinylation experiments, dissociated cerebral cortical neurons were prepared from individual littermates at E17.5. Neurons were cultured in the same way as above at a density of $6.4 \times 10^4$ cells/cm$^2$.

CHO-K1 cells (RCB0285, RIKEN BRC, Tsukuba, Japan) were cultured in HAM's F-12 (Wako Pure Chemical Industries) containing 5% fetal calf serum (FCS) at 37°C in the 5% CO$_2$ incubator. The cells were transfected with plasmids using Lipofectamine 2000 (Thermo Fisher Scientific) in accordance with the manufacturer's protocol. CHO-K1 cells were not included in the list of commonly misidentified cell lines maintained by the International Cell Line Authentication Committee. The origin of the cells (Chinese hamster) was confirmed by PCR in RIKEN BRC (link of datasheet is http://www2.brc. riken.jp/lab/cell/detail.cgi?cell_no=RCB0285). The cells were negative for mycoplasma by both PCR and nuclear staining, which were performed based on protocols by RIKEN BRC (http://cell.brc.riken. jp/ja/quality/myco_kensa).

## Plasmids

To construct the expression vector for Cre, *Cre* cDNA was obtained using RT-PCR from RNA isolated from the cerebral cortex of *Camk2a-Cre* transgenic mouse with primers 5'-GGGGAATTCATG TCCAATTTACTGACC-3' and 5'-CTCGAATTCCTAATCGCCATCTTCCAGC-3'. The product was cloned into the EcoRI site of pEGFP-N1 (Clontech, Mountain View, CA, USA) whose GFP coding sequence was deleted by BamHI/NotI digestion. To construct the expression vector for RNG105 (a.a. 1–122)-GFP, *Rng105* cDNA was obtained by RT-PCR from RNA isolated from mouse cerebral cortex with primers 5'-GTCGACATGCCCTCGGCCACCAGCCACAG-3' and 5'-GAATTCGGAGC TTTATATCTTGACTTAATG-3'. The product was cloned into the XhoI/EcoRI sites of pEGFP-N1 (Clontech). An expression vector for mCherry (pCS2-mCherry) was kindly donated by Dr. N. Kinoshita.

## Western blotting

Extracts of mouse brains were prepared by homogenization in 50 mM Tris (pH 8.0), 150 mM NaCl, 1% NP-40, protease inhibitors (1 mM PMSF, 10 µg/ml leupeptin, pepstatin, and aprotinin) and 1 mM dithiothreitol. After centrifugation for 10 min at 10,000 $\times$ g at 4°C, the supernatant was added to Laemmli sample buffer and boiled. Extracts from cultured CHO cells were prepared in the same way. The extracts were separated by SDS-PAGE, transferred to polyvinylidene fluoride membranes (Merck Millipore, Billerica, MA, USA) and probed with an anti-RNG105 polyclonal antibody (*Shiina et al., 2010*), anti-α-tubulin monoclonal antibody (1:2,000, DM1A, Sigma-Aldrich, St. Louis,

MO, USA), or anti-GFP monoclonal antibody (1:500, GF200, Nacalai Tesque, Kyoto, Japan). Biotinylated secondary antibodies (GE Healthcare, Chicago, IL, USA) and alkaline phosphatase-conjugated streptavidin (GE Healthcare) were used for the detection with a bromochloroindolyl phosphate/nitro blue tetrazolium solution.

## Immunostaining for RNG105

To immunostain brain slices, adult mouse brains were infused with Tissue-Tek (Sakura Finetek, Tokyo, Japan), frozen in liquid nitrogen and sectioned at 10 µm using a cryostat (HM500-OM, Carl Zeiss, Oberkochen, Germany). The sections were mounted on silane-coated coverslips and dried for ~30 min at room temperature. The samples were fixed with 3.7% formaldehyde in phosphate-buffered saline (PBS; 137 mM NaCl, 8.1 mM $Na_2HPO_4$, 1.5 mM $KH_2PO_4$, and 2.7 mM KCl, pH 7.4) for 10 min at room temperature and permeabilized with 0.5% Triton X-100 in PBS. After blocking with 10% FCS in PBS, the samples were incubated with the anti-RNG105 antibody over night at 4°C. After washing with PBS, the samples were incubated with Alexa488-conjugated anti-rabbit IgG (1:400, Jackson ImmunoResearch, West Grove, PA, USA) and 1 µg/ml 4',6-diamidino-2-phenylindole (DAPI) (Wako Pure Chemical Industries) for 1 hr at room temperature to label RNG105 and nuclei, respectively. To immunostain cultured neurons, neurons at 12 DIV were fixed and stained in the same way. Fluorescence images were acquired using an A1 confocal laser microscope equipped with a Ti-E inverted microscope (Nikon, Tokyo, Japan) with a 10 × objective lens or a PlanApo VC60 × oil objective lens.

## Morphological analysis of dendrites and dendritic spines of hippocampal neurons

Brains were removed from the Thy1-GFP expressing *Rng105*[f/f] and *Camk2a-Cre;Rng105*[f/f] mice and fixed with 3.7% formaldehyde in PBS for 2 hr at room temperature. The brains were sectioned at 100 µm using a vibratome VT1200S (LEICA, Wetzlar, Germany), and the sections were mounted in Mowiol (Merck Millipore). Fluorescence images were acquired using the A1 confocal microscope with a 20 × objective lens to measure the length and branching of dendrites, and a 100 × objective lens to measure the size and morphology of spines. The images were analyzed using ImageJ software. Spines were classified as the mushroom type when $W_{neck}/W_{head} < 0.5$ and $W_{head} > 0.4$ µm, where $W_{neck}$ and $W_{head}$ are the width of spine neck and spine head, respectively.

## Glutamate uncaging-induced spine structural LTP

Time-lapse two-photon imaging of dendritic spines was performed using an FVMPE-RS two-photon laser scanning microscope (Olympus, Tokyo, Japan) equipped with an Insight DS Dual-line laser system (Spectra Physics, Santa Clara, CA, USA) and a 95% $O_2$/5% $CO_2$ incubator (Tokai Hit, Shizuoka, Japan) with a water immersion objective lens XLPLN25XWMP2 (Olympus). The culture medium of dissociated hippocampal neurons (12–15 DIV) was exchanged with modified artificial cerebrospinal fluid (ACSF) (125 mM NaCl, 2.5 mM KCl, 3 mM $CaCl_2$, 1.25 mM $NaH_2PO_4$, 26 mM $NaHCO_3$, 20 mM glucose, 1 µM tetrodotoxin [TTX] [Wako Pure Chemical Industries], and 50 µM picrotoxin [Sigma-Aldrich], gassed with 95% $O_2$/5% $CO_2$ before use) containing 2 mM MNI-caged L-glutamate (Tocris, Ellisville, MO, USA). MNI-caged L-glutamate was uncaged locally near single spine heads by 120 pluses (2 ms pulse duration at 2 Hz) of 740 nm laser illumination with laser power of 5 mW. For imaging of spines with mCherry, a 1,040 nm laser was used. 20 images at 0.5 µm focus step were projected by summation, and the fluorescence intensity of spine (sum of pixel intensity in the spine area), which reflects spine volume, was measured using ImageJ software as described previously (*Matsuzaki et al., 2004*).

## Electrophysiology

Hippocampal slice preparation and electrophysiology were performed as described previously (*Shinoda et al., 2011*). Briefly, postnatal 8–10 (P8–10) weeks old *Camk2a-Cre;Rng105*[f/f] mice or *Rng105*[f/f] littermates were deeply anesthetized and decapitated, brains were isolated and cooled rapidly to 4°C. Transverse 400 µm thickness hippocampal slices were prepared using a vibratome RPO7 (D.S.K, Kyoto, Japan) and high-sucrose cutting solution (the formulations of all solutions are described below), and maintained in ACSF at room temperature for at least 2 hr. A bipolar

stimulation electrode was placed in the CA1 SR region to stimulate CA3 Schaffer collateral fibers. Field excitatory postsynaptic potentials (fEPSPs) were recorded using a capillary glass electrode (Harvard Apparatus, Massachusetts, MA, USA) filled with ACSF placed in the CA1 SR after a 0.05 Hz test pulse generated by a pulse generator Master-8 (A.M.P.I., Jerusalem, Israel) equipped with an isolator ISO-Flex (A.M.P.I.). For LTP recording and theta-burst stimulation, 50% maximum stimulus intensity was used. LTP was induced by theta-burst stimulation (Four trains with 10 s intervals between trains; each train had five bursts separated by 200 ms and included four pulses delivered at 100 Hz). Data were amplified using a MultiClamp 700A (Molecular Devices, Sunnyvale, CA, USA), digitized at 10 kHz and filtered at 2 kHz using a Digidata 1440 system (Molecular Devices) with pCLAMP9 software (Molecular Devices). The formulation of the solutions were (in mM): High-sucrose cutting solution; 234 sucrose, 2.5 KCl, 1.25 $NaH_2PO_4$, 0.5 $CaCl_2$, 10 $MgSO_4$, 26 $NaHCO_3$, and 11 D-glucose, gassed with 95% $O_2$/5% $CO_2$. ACSF; 125 NaCl, 2.5 KCl, 1.25 $NaH_2PO_4$, 2 $CaCl_2$, 1 $MgCl_2$, 26 $NaHCO_3$, and 11 D-glucose, gassed with 95% $O_2$/5% $CO_2$.

## Open field test

In all behavioral tests, male mice (3–4 months old; 2–4 mice were co-housed in a 12 hr dark/light cycle) were used during the light cycle. The open field test was conducted in a round apparatus (85 cm in diameter with a 40 cm wall) with concentric circles and radial lines drawn on the floor, which divided the floor into 25 blocks. A mouse, naive to the apparatus, was placed in the center of the field and allowed to explore for 5 min. The total number of line crossings was recorded as an index of exploratory activity. The mouse was subjected to the test on three consecutive days to assess habituation to the novel place.

## Light/dark transition test

The test was conducted with an apparatus consisted of a light chamber and a dark chamber separated by a sliding door (LDK-M, Melquest, Toyama, Japan). A mouse was automatically monitored using a SCANET system (Melquest). The mouse was placed into the light chamber, and 5 s later, the door was opened. The mouse was allowed to move freely between the two chambers for 5 min. Distance traveled, time spent in each chamber, and the number of transition between the chambers were measured.

## Novel object recognition test

To habituate to the test environment, mice were placed in a chamber (43 × 43 × 29.5 cm) and allowed to explore freely for 1.5 hr on four consecutive days. In the first session, two identical objects (object 1 and object 2, cell culture flasks) were placed at the opposite corners of the chamber, and the mouse was allowed to explore the chamber for 5 min. In the second session, object two was replaced by a different type of object (object 3, a wooden prism), and the mouse was allowed to explore the chamber for 5 min. The interval between the sessions was 2.5 hr. The number of mouse interactions with the objects was counted, where object interaction was defined as the nose directing toward the object at a distance ≤2 cm.

## Rotarod test

The rotarod test was conducted using ROTA-ROD for mice 7650 (Ugo Basile, Varese, Italy). A mouse was placed on a rod rotating at a fixed speed of 24 rpm. The duration of a trial was 3 min, and if the mouse fell from the rod within 3 min, the mouse was re-placed on the rod. The number of falls and the longest latency without falling within a trial were measured. The mouse was subjected to the test on three consecutive days to assess motor skill learning.

## Morris water maze

The Morris water maze was conducted in a round water tank with a diameter of 110 cm filled with ~24°C water. First, the mouse was subjected to visible platform test. In this test, a black platform (6 cm in diameter) was placed 0.5 cm above the water surface. A mouse was placed in the pool and the escape latency to the platform was measured. If the mouse found the platform within a 1 min time limit, the mouse was allowed to stay on the platform for 30 s. If not, the mouse was guided to the platform before staying on the platform for 30 s. Mice were given six trials per day for five

consecutive days. The location of the platform and the start position were changed randomly in each trial.

Next, the mice were subjected to a hidden platform test, in which a clear platform (6 cm in diameter) was placed 1 cm below the water surface. The water was clouded with non-toxic white paint (Sakura Color Products, Osaka, Japan). Spatial cues (four different plane figures) were attached on the interior wall of the tank above the water surface. The test was conducted in the same way as in the visible platform test with a 2 min time limit. The start position was changed randomly, but the location of the platform was fixed. The mice were given six trials per day for 10 consecutive days.

One day after the last trial of the hidden platform task, the mice were subjected to a probe test. The platform was removed from the pool and the swimming path of the mouse was tracked for 1 min using a computer-based video tracking system ANY-Maze (Stoeling, Wood Dale, IL, USA). Time spent in the target quadrant, where the hidden platform had been placed, and in the other quadrants was also measured.

## Contextual fear conditioning test (passive avoidance)

The test was conducted using the LDK-M chamber (Melquest) with the SCANET system (Melquest). One day before the training day, to habituate to the test environment, the mouse was allowed to freely explore both chambers for 30 min with the door opened. On the training day, the mouse was placed in the light chamber and allowed to freely explore the chambers for 15 min after the door opened, and the time spent in the dark chamber during the first 5 min was measured (Pre-FS). After that, when the mouse entered the dark chamber, the door was closed and foot shock (0.5 mA, 1 s duration, 4 times with 1 min intervals) was delivered. After the foot shock, the mouse was returned to its home cage. The mouse was replaced in the light chamber at 5 min, 1 day, and 1 week after the foot shock and allowed to freely explore the chambers for 5 min after the door opened, and the time spent in the dark chamber was measured.

## Contextual fear conditioning test (context freezing)

A mouse was placed in a test chamber (15 × 15 × 15 cm, the dark chamber of LDK-M with the door closed) and allowed to explore freely for 30 s. Then the mouse received a foot shock (0.5 mA, 2 s) and was returned to its home cage. Five days after the conditioning, the mouse was replaced in the test chamber and in a control chamber (24.5 × 15.5 × 14.8 cm, CL-0112–3, CLEA Japan, Tokyo, Japan) for 1 min each. Freezing time in the chambers was measured by ANY-Maze.

## Tissue dissection and isolation of SP and SR from hippocampus

Mouse brains (P12 weeks old) were removed in ice cold sterile PBS, and embedded in a 4.5% low-melting point agarose (Agarose XP, Nippon Gene, Tokyo, Japan). Coronal sections (500 μm thick) were sliced in ice-cold sterile PBS using the vibratome VT1200S, and transferred into ice-cold sterile PBS. Brain slices were stained with 1 μg/mL DAPI for 10 min at 4°C. Stratum pyramidale (SP) and stratum radiatum (SR) in the hippocampal CA1 region were microdissected manually using glass capillaries (G-1, Narishige, Tokyo, Japan) with the use of a stereomicroscope. Glass capillaries were made using a needle puller (model PB-7, Narishige). The borders between the stratum oriens (SO) and SP, and between the SP and SR were cut to isolate the SP, and the border between SP and SR, and between SR and stratum lacnosum-moleculare (SLM) were cut to isolate the SR. To avoid contamination of SR with the soma of pyramidal neurons, the isolated SR was checked to confirm it did not contain a high-density DAPI-positive nucleic layer (SP) with the use of an inverted fluorescence microscope (IX83, Olympus) with a 40 × objective lens. To obtain sufficient amount of RNA for RNA-seq, left and right hippocampi from three mice were dissected and collected into one sample tube. Triplicate samples were prepared for RNA-seq analysis. The samples were stored at −80°C until RNA extraction.

## RNA extraction

Total RNA was extracted from the tissues (SP and SR) using ISOGEN (Nippon gene) in accordance with the manufacturer's protocols with minor modifications. The isolated tissue was homogenized in 400 μL ISOGEN by pipetting. After the homogenate was stored at room temperature for 5 min, 100 μL chloroform was added and the sample was shaken vigorously for 30 s. After being stored on ice

for 5 min, the sample was centrifuged for 15 min (21,900 × g, 4°C), and about 240 µL of the aqueous phase containing RNA was collected in a new tube. The sample was centrifuged for 15 min (21,900 × g, 4°C) again and the aqueous phase was collected into a new tube. To remove phenol and chloroform from the sample, 240 µL diethyl ether was added to the sample and vortexed for 1 min. After the sample was centrifuged for 1 min (21,900 × g, 4°C), the upper phase was removed. This diethyl ether treatment was performed three times. Then, 25 µL 3 M sodium acetate and 250 µL 2-propanol were added to the sample and mixed, and then RNA was precipitated on ice for 30 min. After the sample was centrifuged for 15 min (21,900 × g, 4°C), the supernatant was removed. The RNA precipitate was washed with 500 µL 70% ethanol, dried using a vacuum desiccator and dissolved in 22 µL RNase-free water (TaKaRa, Shiga, Japan).

Next, DNA was removed from the sample by DNase treatment. 0.5 µL DNase (RT grade) (Nippon Gene) and 10 × buffer were added to the 22 µL RNA solution, and the sample was incubated for 15 min at 37°C. Then, 25 µL phenol-chloroform mixture was added, and the sample was vortexed for 30 s. After the sample was centrifuged for 10 min (21,900 × g, 4°C), upper phase was collected. Phenol and chloroform were removed from the sample by diethyl ether treatment as described above. Then, 2.5 µL 3 M sodium acetate and 62.5 µL 100% ethanol were added to the sample and mixed, and then RNA was precipitated at −20°C for 30 min. After the sample was centrifuged for 20 min (21,900 × g, 4°C), the supernatant was removed. The RNA precipitate was washed with 150 µL 70% ethanol, dried, and dissolved in 10 µL RNase-free water. RNA solution was stored at −80°C until use for the preparation of cDNA libraries.

## Preparation of cDNA libraries for RNA-seq

Before preparation of cDNA libraries, the quality of the extracted RNA was checked. RNA integrity was measured using an Agilent 2100 bioanalyzer (Agilent Technologies, Santa Clara, CA, USA) with RNA 6000 Nano Kit and RNA 6000 Pico Kit (Agilent Technologies) in accordance with the manufacturer's protocols. The RNA integrity number was over seven for all samples, which was sufficient quality for RNA-seq.

Total RNA (200 ng per sample) was used as the starting material to prepare cDNA using TruSeq RNA sample Preparation Kit v2 (Illumina, San Diego, CA, USA) at a half scale compared with the manufacturer's protocols, as follows. Poly-A-containing mRNAs were purified using Oligo dT magnetic beads. After the mRNAs were denatured, they were fragmented at 94°C for 4 min. First-strand cDNA was synthesized using SuperScript II Reverse Transcriptase (Thermo Fisher Scientific), and then second-strand cDNA was synthesized. After the ends of the fragments were blunted, an adenine nucleotide was added to the 3'-end at 37°C for 30 min. RNA adapter index was ligated to the 5'- and 3'-ends of the ds cDNA, and the cDNA was amplified with 10 PCR cycles. cDNA quality was validated using the bioanalyzer with a High Sensitivity DNA Kit (Agilent Technologies). The cDNA libraries were quantified with quantitative PCR (qPCR) using a 7500 real-time PCR system (Thermo Fisher Scientific) and adjusted to 2 nM.

## Next-generation RNA-seq and read alignment

In each experiment, an equal amount of cDNA from SP and SR was analyzed, and three independent experiments were conducted. Twelve cDNA libraries (four kinds of samples × three biological replicates) were sequenced with 101 bp paired-end sequencing using a HiSeq1500 (Illumina). The resulting reads were mapped to the mouse genome (*Mus musculus* Ensemble NCBIM37) using TopHat (v 2.0.11). The mapped reads were assembled using Cufflinks (v 2.2.1), and differential gene enrichment analysis was conducted using Cuffdiff.

## Differential mRNA enrichment analysis of RNA-seq data

S-mRNAs and D-mRNAs in control (*Rng105*^f/f) mice were identified as follows. First, mRNAs from SP and SR of *Rng105*^f/f mice were analyzed using Cuffdiff, as described above, which identified mRNAs whose concentration (FPKM value) was significantly different between SP and SR, designated as 'yes', and not significantly different between the layers designated as 'no' in the 'all mRNAs' list (*Supplementary file 1*). After this, mRNAs, whose FPKM values were more than zero both in SP and SR, were selected and subjected to subsequent data analysis. 'Yes' mRNAs were divided into two groups, candidates for dendritic mRNAs whose DAIs (relative FPKM value in SR to SP) were more

than 1, and candidates for somatic mRNAs whose DAIs were less than 1. Next, to identify mRNAs specifically expressed in pyramidal neurons, we eliminated mRNAs also expressed in other types of cells such as glial cells, interneurons, and endothelial cells (*Cajigas et al., 2012*; *Doyle et al., 2008*; *Daneman et al., 2010*; *Cahoy et al., 2008*) from each of the lists (*Supplementary file 2A,B*). Finally, we identified 1122 dendritic mRNAs, 2106 somatic mRNAs, and 2814 non-significant mRNAs in *Rng105*$^{f/f}$ mice (*Supplementary file 3A−C*). Furthermore, to compare the dendritic localization of mRNAs between *Rng105*$^{f/f}$ and *Camk2a-Cre;Rng105*$^{f/f}$ mice, the ratio of DAI in *Camk2a-Cre; Rng105*$^{f/f}$ mice to *Rng105*$^{f/f}$ mice was calculated for each mRNA and indicated in the 'DAI (*Camk2a-Cre;Rng105*$^{f/f}$)/DAI (*Rng105*$^{f/f}$)' columns (*Supplementary file 3A−C*).

## Gene ontology analysis of RNA-seq data

Gene ontology enrichment analysis was performed using DAVID functional annotation tools. Significance of overrepresentation of GO terms was assessed using the Benjamini-Hochberg false discovery rate (FDR) criterion at $p < 0.05$.

## Homeostatic scaling of GluR1 and GluR2 surface expression on dendrites

Immunostaining of cultured cortical neurons (9 DIV) for GluR1 and GluR2 was conducted as described previously (*Ohashi et al., 2016*). To block neuronal activity, 0.7 µM TTX and 20 µM D-2-amino-5-phosphonovaleric acid (APV) (Sigma-Aldrich) were added to the medium for 24 hr at 37°C in a 5% $CO_2$ incubator. Live neurons were incubated with an anti-GluR1 (1:15, PC246, Merck Millipore) or an anti-GluR2 antibody (1:100, MAB397, Merck Millipore) at 37°C in a 5% $CO_2$ incubator for 1 hr. After the neurons were washed in Neurobasal-A medium, they were fixed with 3.7% paraformaldehyde in PBS for 20 min at 25°C. Fixed neurons were blocked for 30 min in 10% FCS in DMEM (Sigma-Aldrich), and incubated with an Alexa Fluor 488-conjugated anti-rabbit and mouse IgG antibodies (1:400, Thermo Fisher Scientific) in 10% FCS in DMEM for 3 hr at 25°C to label cell surface GluR1 and GluR2. After neurons were washed in PBS, they were fixed again and permeabilized with 0.25% Triton X-100 in PBS for 10 min. After the neurons were blocked, they were incubated with the anti-GluR1 antibody (1:50) or anti-GluR2 antibody (1:100) for 12 hr at 4°C and then with a Cy3-conjugated anti-rabbit and mouse IgG antibodies (1:400, Jackson Immuno Research) for 3 hr at 25°C to label intracellular and residual surface GluR1 and GluR2.

Neurons were imaged using the IX83 inverted fluorescence microscope (Olympus) with a 40 × objective lens and an ORCA-R2 digital CCD camera (Hamamatsu Photonics, Hamamatsu, Japan). Because GluR1 and GluR2 were detected in a punctate manner, they were quantified by counting the number and measuring the fluorescence intensity of the puncta in dendrites. To count the number of GluR1 and GluR2 puncta, the images were converted into binary images by selecting dendritic regions and using the MaxEntropy threshold algorithm in ImageJ software. The number of GluR1 and GluR2 puncta in dendrites was counted using the magic wand tool and analysis tool in Adobe Photoshop software. To count total GluR1 and GluR2 puncta, the binary images of before and after permeabilization were merged and used. To measure the fluorescence intensity of GluR1 and GluR2 puncta in dendrites, the binary image and original image were layered in Adobe Photoshop software. ROIs were selected in the binary layer using the magic wand tool, and fluorescence intensity in the ROIs in the original layer was calculated by multiplying mean pixel intensity by area of the ROIs. The sum of fluorescence intensity was normalized by dendrite length and by the intensity of GluR1 and GluR2 in the soma. GluR1 and GluR2 intensity in the soma was calculated using the layered images by multiplying mean pixel intensity by area of ROIs in the soma and normalized by the area of the soma. Fluorescence intensity of GluR1 and GluR2 puncta immunostained before permeabilization was normalized by that immunostained after permeabilization.

## Biotinylation assay for surface AMPARs

Biotinylation assay was conducted as described previously (*Chung et al., 2000*; *Snyder et al., 2001*; *Aoto et al., 2008*). Primary cultured cortical neurons (9 DIV) in two 30 mm dishes were treated with or without 1 µM TTX and 100 µM APV in the culture medium for 24 hr prior to surface biotinylation (*Aoto et al., 2008*). The dishes were placed on ice and washed three times with ice-cold ACSF (124 mM NaCl, 5 mM KCl, 1.25 mM $NaH_2PO_4$, 26 mM $NaHCO_3$, 0.8 mM $MgCl_2$, 1.8 mM $CaCl_2$, and 10

mM D-glucose, gassed with 95% $O_2$/5% $CO_2$). Then the neurons were incubated with ACSF containing 1 mg/ml sulfo-NHS-LC biotin (Thermo Fisher Scientific) for 30 min on ice. After washing once with ice-cold 100 mM glycine in ACSF and three times with ice-cold TBS (50 mM Tris, 150 mM NaCl, pH 7.5), the neurons were lysed in 100 µl of modified RIPA buffer (1% Triton X-100, 0.1% SDS, 0.5% deoxycholic acid, 50 mM $NaHPO_4$ [pH 7.2], 150 mM NaCl, 2 mM EDTA, 25 mM β-glycerophosphate, 1 mM PMSF, 10 µg/ml leupeptin). The lysate was centrifuged at 14,000 × g for 15 min at 4℃, and 85 µl of supernatant was incubated with 40 µl of NeutraAvidin Agarose beads (Thermo Fisher Scientific) for 3 hr at 4℃ with gentle rocking. After washing three times with modified RIPA buffer, biotinylated proteins were eluted from the beads with 80 µl of SDS sample buffer and boiled for 5 min. The total lysate and biotinylated eluate were analyzed by western blotting with the anti-GluR1 (1:50, PC246, Merck Millipore), and anti-GluR2 (1:1000, MAB397, Merck Millipore) antibodies. Alkaline phosphatase-conjugated secondary antibodies (1:5000, 711-055-152 and 115-055-146, Jackson Immuno Research) and Can Get Immunoreaction Enhancer Solution (TOYOBO, Osaka, Japan) were used for the detection with a bromochloroindolyl phosphate/nitro blue tetrazolium solution. Quantification of the band intensity was conducted as previously described (*Ohashi et al., 2016*). A standard dilution series of the lysate from cultured neurons was loaded on the same gel, and used to generate a standard curve and calculate the relative intensity of upper and lower bands of GluR1 and GluR2. The band intensity was measured using ImageJ software. Triplicate from three mice (nine samples) were analyzed for each group.

## Quantification and statistical analysis

Sample numbers and experimental repeats are indicated in figure legends. Statistical significance was determined using Student's t-test, paired t-test, one-way ANOVA, one-way repeated measures ANOVA, two-way ANOVA, two-way repeated measures ANOVA, post-hoc Tukey-Kramer test, and Bonferroni post-hoc t-test as indicated in the figure legends. Statistical analysis was performed in R, Excel, or an Excel add-in software Statcel (The Publisher OMS Ltd., Saitama, Japan). Exact F, t, and p-values are indicated in Statistical reporting table (*Supplementary file 7*). Post-hoc power analysis was performed with G*Power (http://www.gpower.hhu.de/) and statistical power is indicated in the Statistical reporting table. No blinding method was used in this study. In behavioral tests, animals that died before the experimental endpoint were excluded from the data analysis. The Gene ontology term enrichment was analyzed using DAVID functional annotation tools (https://david.ncifcrf.gov/).

## Data resources

Raw and processed data files for the RNA-seq analysis have been deposited in the NCBI Gene Expression Omnibus (GEO) under series accession number GSE96552 (https://www.ncbi.nlm.nih.gov/geo/query/acc.cgi?acc=GSE96552).

## Acknowledgements

We thank Dr. M Matsuzaki for advice on glutamate uncaging experiments; C Matsuda for technical assistance; Spectrography and Bioimaging Facility and Functional Genomic Facility, NIBB Core Research Facilities for technical supports. This work was supported by Grant-in-Aid for Scientific Research on Priority Areas (Molecular Brain Science) from the MEXT of Japan, JSPS KAKENHI Grant Number 16K07361 to NS, and JSPS KAKENHI Grant Number 16J07985 to RO (RO is a JSPS Research Fellow). The work was also supported by the Uehara Memorial, the DAIKO, and the Toyoaki foundations to NS.

## Additional information

### Funding

| Funder | Grant reference number | Author |
|---|---|---|
| Ministry of Education, Culture, Sports, Science and Technology | Grant-in-Aid for Scientific Research on Priority Areas (Molecular Brain Science) | Kenji Sakimura Nobuyuki Shiina |

| Japan Society for the Promotion of Science | KAKENHI Grant Number 16K07361 | Nobuyuki Shiina |
| Japan Society for the Promotion of Science | KAKENHI Grant Number 16J07985 | Rie Ohashi |
| Uehara Memorial Foundation | | Nobuyuki Shiina |
| Daiko Foundation | | Nobuyuki Shiina |
| Toyoaki Foundation | | Nobuyuki Shiina |

The funders had no role in study design, data collection and interpretation, or the decision to submit the work for publication.

## Author contributions

Kei Nakayama, Conceptualization, Formal analysis, Investigation, Writing—original draft; Rie Ohashi, Conceptualization, Formal analysis, Funding acquisition, Investigation, Writing—original draft; Yo Shinoda, Formal analysis, Investigation, Writing—original draft; Maya Yamazaki, Manabu Abe, Resources, Investigation; Akihiro Fujikawa, Akira Futatsugi, Masaharu Noda, Katsuhiko Mikoshiba, Teiichi Furuichi, Resources, Writing—review and editing; Shuji Shigenobu, Kenji Sakimura, Resources; Nobuyuki Shiina, Conceptualization, Supervision, Funding acquisition, Writing—original draft, Project administration, Writing—review and editing

## Author ORCIDs

Yo Shinoda http://orcid.org/0000-0003-3253-2194
Katsuhiko Mikoshiba https://orcid.org/0000-0002-3487-6970
Nobuyuki Shiina http://orcid.org/0000-0002-1854-4239

## Ethics

Animal experimentation: All animal care, experiments and behavioral testing procedures were approved by the Institutional Animal Care and Use Committee of the National Institutes of Natural Sciences (Permit Number: 17A059), and performed in accordance with the guidelines from the National Institutes of Natural Sciences, Niigata University and the Science Council of Japan.

## Decision letter and Author response

Decision letter https://doi.org/10.7554/eLife.29677.028
Author response https://doi.org/10.7554/eLife.29677.029

# Additional files

## Supplementary files

• Supplementary file 1. Differential mRNA enrichment analysis between SP and SR of hippocampus − All mRNAs (36,701)
DOI: https://doi.org/10.7554/eLife.29677.018

• Supplementary file 2. (**A**) mRNAs eliminated from the candidates for dendritic mRNAs. (**B**) mRNAs eliminated from the candidates for somatic mRNAs
DOI: https://doi.org/10.7554/eLife.29677.019

• Supplementary file 3. (**A**) Dendritically enriched mRNAs (1,122). (**B**) Somatically enriched mRNAs (2,106). (**C**) mRNAs not significantly enriched in dendrites nor soma (2,814)
DOI: https://doi.org/10.7554/eLife.29677.020

• Supplementary file 4. (**A**) Dendritic mRNAs classified in 'regulation of Arf protein signal transduction'. (**B**) Dendritic mRNAs classified in 'structural constituent of ribosome'. (**C**) Dendritic mRNAs classified in 'translation'. (**D**) Dendritic mRNAs classified in 'Fc gamma R-mediated phagocytosis'. (**E**) Dendritic mRNAs classified in 'pleckstrin homology'. (**F**) Dendritic mRNAs classified in 'chemotaxis'. (**G**) Dendritic mRNAs classified in 'GTPase regulator activity'. (**H**) Dendritic mRNAs classified in 'leukocyte activation'. (**I**) Dendritic mRNAs classified in 'SH3'
DOI: https://doi.org/10.7554/eLife.29677.021

• Supplementary file 5. (**A**) mRNAs for AMPA receptor subunits and dendritic mRNAs for AMPA receptor regulators. (**B**) SR-enriched mRNAs involved in membrane potential regulation
DOI: https://doi.org/10.7554/eLife.29677.022

• Supplementary file 6. mRNAs reduced in *Camk2a-Cre;Rng105*^f/f soma
DOI: https://doi.org/10.7554/eLife.29677.023

• Supplementary file 7. Statistical reporting table
DOI: https://doi.org/10.7554/eLife.29677.024

• Transparent reporting form
DOI: https://doi.org/10.7554/eLife.29677.025

## Major datasets

The following dataset was generated:

| Author(s) | Year | Dataset title | Dataset URL | Database, license, and accessibility information |
|---|---|---|---|---|
| Ohashi R, Shigenobu S, Shiina N | 2017 | Genome-wide profiling of somato-dendritic mRNA distribution in the hippocampus in RNG105/caprin1 conditional knockout (cKO) mice | https://www.ncbi.nlm.nih.gov/geo/query/acc.cgi?acc=GSE96552 | Publicly available at the NCBI Gene Expression Omnibus (accession no: GSE96552) |

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
