## [Decision Letter]

Thank you for submitting your article "RNG105, an RNA granule protein for mRNA localization, is essential for long-term memory formation" for consideration by *eLife*. Your article has been reviewed by three peer reviewers, one of whom, Mani Ramaswami, also served as the Reviewing Editor. The evaluation was overseen by a Senior Editor.

The reviewers have discussed the reviews with one another and the Reviewing Editor has drafted this decision to help you prepare a revised submission

Summary:

The authors create and analyze mouse mutants deficient in the RNG105/ Caprin1 protein, which has previously been shown to be a shared component of stress granules and neuronal RNA granules. The work examines the histological, electrophysiological, and cognitive impact of the selective perturbation of RNG105 gene from the forebrain. The study uses cell culture, ex vivo slice preparations, and in vivo murine behavioral studies to approach the experimental questions of the study. Because RNG105 is involved in mRNA transport, the study concludes with a comprehensive characterization of mRNA identification in somato-dendritic compartments. To do this, the authors isolated mRNAs from stratum pyramidale (SP) and stratum radiatum (SR) and then performed RNA-Seq analysis on the isolated mRNAs. These molecular analyses constitute the clearest and most exciting features of this study, providing direct and compelling evidence that RNG105-mediates the intracellular localization specific cohorts of mRNA.

Additional arguments and observations link this function of Caprin1 to long-term memory (LTM). Loss of RNG105 also results in significant changes to dendritic spine formation, reduced excitatory synaptic transmission, and robust LTM deficits. The authors also observe that in their final experiment that activity regulated AMPA receptor trafficking is specifically perturbed in the Caprin1 mutant mouse.

Overall the study is interesting, providing data potentially linking Caprin1 function to mRNA transport and providing evidence for a causal role of such transport in neural plasticity and LTM. However, the analysis and discussion of several experiments and issues would benefit from increased depth and rigor. Several issues need to be addressed before it is ready for publication.

1) The conditional allele is not a null. Thus, it should not be referred to as a conditional KO mouse, but rather as conditional exon 5-6 deletion. This is not only a didactic point, but also one that should be explicitly acknowledged in the interpretation of the data. This is also significant because heterozygous nulls can have phenotypes – so perhaps a homozygous weak allele mimics the heterozygous null? It would be useful if the authors briefly explained why they did not create a conditional null (a complete ORF deletion) in the first place.

2) The text should mention what the presumed function may be of the protein-synthesis dependent dendritic volume expansion observed after 15 minutes of stimulation? It is important to explain how and whether this may relate to memory formation, particularly if this process is not required for normal late-LTP.

3) The electrophysiological results presented in Figure 4 have some problematic interpretations.

First, the Results section states that no PPD phenotype for KO was observed yet, significant results are reported in Figure 4 (which appear substantiated by both the traces in 4C and the data in 4D). Can the authors please clarify this?

Second, it is difficult compare LTP across preparations where input/output (I/O) differ. Because the KO mice have ~50% reduction in response and LTP is a direct function of the input, the authors need to correct for differences in I/O comparing LTP across genotypes.

There is no description provided here of the input strength (% of max fEPSP) used for tetanic brain stimulation (TBS). Typically, the stimulus is delivered at 30-50% of the maximum fEPSP for each genotype, it is unclear what was used here. The graph in Figure 4 shows comparable LTP induction post-TBS between WT and KO. There is some text in the last paragraph of the subsection “RNG105 cKO reduces synaptic responses to stimulation” that seems to address the confound here but, it is unclear how the data was processed to arrive at the graph seen in 4F. Does the Y-axis label "% of baseline" refer to the percent of genotype baseline? If so, then this is not ideal because the input/output responses from KO slices is reduced. The convention is to show% of baseline control, which in this case is the WT littermate. This should be clarified because it is very likely that the KO mice do exhibit LTP abnormalities that are masked by this non-standard analysis. Theta burst is a very strong LTM-inducing protocol so these experiments could be masking differences that may be observed with weaker LTP-inducing protocols. If an alternative conclusion is reached, then it could resolve the somewhat confusing dichotomy between observations on late LTP (normal) and "long-term memory," (defective) in RNG105 mutants.

4) There are some caveats to the interpretation of behavioral and memory tests that should be addressed.

a) The finding that motor learning is normal in RNG105 mutants is consistent with the finding that conditional excision of exon-5 and 6 is less efficient in the cerebellum. But it is equally consistent with the idea that RNG105 is simply not required for this form of learning and memory. This should be acknowledged.

b) The finding that RNG105 mutants show defects in habituation to their environments is interesting. However, the data in Figure 5 should be analyzed by RM-ANOVA not, single trial ANOVA. Could the results be explained by greater activity (increased distance moved) in the cKO rather than a habituation defect? Habituation is better assayed using other tests (EPM, Object Location, Object recognition) and any conclusion on habituation should be strengthened by additional experimentation or moderated by acknowledging caveats.

If the authors consider it reasonable, then the Discussion should consider whether increased stress and anxiety could inhibit habituation particularly given that mutations in Caprin 1 are associated with autism.

c) The inability of RNG105 mutants to find the hidden platform in the Morris Maze experiment is confusing. The performance in Figure 6 suggests the mice are incapable of learning the task, making the probe test somewhat pointless (floor effect). Was outlier analysis performed on these data sets? The variance in the KO mice is quite high.

5) The analysis of mRNAs and pathways using GO approaches and subsequent selection of pathways to comment on is typical and not unreasonable, but the discussion and analyses should be expanded to clearly acknowledge alternative unknown and possibly indirect mechanisms by which RNG105 mutations could result in phenotypes that are described.

6) There appears to be a section missing in the Results section. It appears that Figure 9 is not referred to in the text? If so, then this figure should be removed or the text for it should be added.

7) The premise behind Figure 10 is interesting but the approach used is highly variable and subject to many experimental factors (i.e. staining robustness) that can make this type of analysis challenging especially across multiple cultures. One would be much more confident of this result if it was supported by a more quantitative approach such as biotin surface labeling and immunoblot measurement of labeled proteins. Normalization is much more reliable with this approach. Additionally, the Materials and methods section should be updated to include more information about how ROIs are used and sized, whether 'n' refers to independent cultures or cells in a single culture etc. This is the only mechanistic experiment in the study and the results have important implications for RNG105 function and should ideally be validated using another approach. Finally, there are four groups here and student's t-test is not the right test for this experiment, the variability is very high in the samples and overall effects modest, it would be important to see if significant findings hold up to proper posthoc testing.

8) The paper ends with the observation that "Thus, this study provides… new knowledge on a physiological function of RNA granules." This is not true at all – as the effect of caprin mutations on granules is never studied, only the function of the Caprin1 protein is analyzed, which could be distinct from the function of an RNA granule.

9) The authors correctly identify a major departure from their findings from similar studies that also examined RNA constituents but did not find major impairments in LTM. However, the authors do not attempt to speculate on why this might be. This is especially important because some of these previous studies examine RNA granule components such as Ataxin-2 that have actually been shown to be required to RNA granule formation. Some discussion to this point would be helpful. Also, there are several papers analysing functions of other granule proteins (e.g. Gld-2, Atx2, Staufen, Pumilio, DDX6/Me31B) on long-term memory that are not cited or discussed, in particular from *Drosophila* work.

10) FMRP mutants in *Drosophila* have been reported to have specific defects in long-term memory. Could the different observation in mammals be due to the existence of alternative isoforms? How many isoforms of RNG105 and G3BP exist in rodents and how much could or do their functions overlap in vivo? This issue should be broadly considered in the Discussion as moderating caveat to the mutant analysis described.

---

## [Author Response]

[…] Overall the study is interesting, providing data potentially linking Caprin1 function to mRNA transport and providing evidence for a causal role of such transport in neural plasticity and LTM. However, the analysis and discussion of several experiments and issues would benefit from increased depth and rigor. Several issues need to be addressed before it is ready for publication.1) The conditional allele is not a null. Thus, it should not be referred to as a conditional KO mouse, but rather as conditional exon 5-6 deletion. This is not only a didactic point, but also one that should be explicitly acknowledged in the interpretation of the data. This is also significant because heterozygous nulls can have phenotypes – so perhaps a homozygous weak allele mimics the heterozygous null? It would be useful if the authors briefly explained why they did not create a conditional null (a complete ORF deletion) in the first place.

Thank you for this comment. RNG105 cKO mice have been referred to as *Camk2a-Cre;Rng105*^f/f^ mice in the revised manuscript. In addition, "knockout" has been changed to "deletion". It is technically impossible to produce a recombinant clone deleting all coding exons, just with a one-time homologous recombination, when the length between two loxPs is too long. To generate RNG105/caprin1 conditional deletion mice, we deleted exons 5−6 alone. However, we consider this deletion would be sufficient to produce cKO mice because of the following reasons. In *Camk2a-Cre;Rng105*^f/f^ mice, frame shift occurs in the exon 5−6-deleted mRNA, which will produce truncated RNG105 protein lacking most of the functional domains ranging from the N-terminal coiled-coil domain to the C-terminal RG-rich domain (a.a. 123−707) (Shiina et al., 2005). In addition, this kind of mRNAs harboring nonsense mutation become unstable and degraded by nonsense-mediated mRNA decay (NMD), resulting in hardly any production of truncated peptide. In fact, in the hippocampus where Cre was highly expressed, expression of all *Rng105* exon transcripts was reduced to the comparable level to that of exon 5−6 transcripts as judged by RNA-seq analysis (Figure 1—figure supplement 1). Furthermore, a polyclonal anti-RNG105 antibody, which recognized truncated RNG105 protein encoded by exons 1−4, did not detect any truncated form of RNG105 in the cerebrum of *Camk2a-Cre;Rng105*^f/f^ mice (Figure 1—figure supplement 1). We have added these new supplement data and descriptions in the text (subsection “Generation of RNG105 conditional deletion mice”, first and last paragraphs). These results supported the notion that *Camk2a-Cre;Rng105*^f/f^ mice are cKO mice. Consequently, *Camk2a-Cre;Rng105*^f/f^ was not a weak allele, as shown in the behavioral analyses: *Camk2a-Cre;Rng105*^f/f^ mice showed much severer deficits in spatial memory than RNG105 heterozygous null mice which showed normal spatial memory except for reversal learning (Ohashi et al., 2016).

2) The text should mention what the presumed function may be of the protein-synthesis dependent dendritic volume expansion observed after 15 minutes of stimulation? It is important to explain how and whether this may relate to memory formation, particularly if this process is not required for normal late-LTP.

Dendritic spine volume is tightly correlated with AMPAR expression level and EPSC in the spine (Kopec et al., J. Neurosci., 2007; Matsuzaki et al., Nat. Neurosci., 2001). In addition, spine volume expansion is dependent on translation. Therefore, spine volume expansion is believed to be associated with translation-dependent late-phase LTP and long-term memory formation. However, in the present study, we observed normal late-phase LTP without spine volume expansion. From these results, we have modified the accepted notion and proposed a hypothesis that spine volume expansion may increase the limit of absolute amplitude of EPSP. This hypothesis is described in the following paragraph: "Another, though not mutually exclusive, explanation is that translation deficiency may not necessarily cause the decline of LTP in the late phase, but limit the absolute amplitude of EPSP to a certain level in the late phase of LTP. […] *As a result, LTP may appear to be sustained in the late phase, but because of the low absolute amplitude of EPSP, long-term memory may be affected.*"

3) The electrophysiological results presented in Figure 4 have some problematic interpretations.First, the Results section states that no PPD phenotype for KO was observed yet, significant results are reported in Figure 4 (which appear substantiated by both the traces in 4C and the data in 4D). Can the authors please clarify this?

We are sorry for the confusing description. In Figure 4, significant difference was found between *Camk2a-Cre;Rng105*^f/f^ and *Camk2a-Cre;Rng105*^f/f^ (LTP). There was no significant difference between *Rng105*^f/f^ and *Camk2a-Cre;Rng105*^f/f^. We have added the sentence "There was no significant difference between the genotypes" in the Figure 4 legend. In addition, we have added the description "Paired-pulse ratio (PPR) was significantly decreased after LTP induction in Camk2a-Cre;Rng105^f/f^ mice, but it was not significantly different between the genotypes (Figure 4)" to clarify that the significant difference was not between the genotypes but between *Camk2a-Cre;Rng105*^f/f^ and *Camk2a-Cre;Rng105*^f/f^ (LTP).

Second, it is difficult compare LTP across preparations where input/output (I/O) differ. Because the KO mice have ~50% reduction in response and LTP is a direct function of the input, the authors need to correct for differences in I/O comparing LTP across genotypes.

We are grateful for this comment. Because the I/O response in *Camk2a-Cre;Rng105*^f/f^ mice was reduced to 45.1% of that in *Rng105*^f/f^ mice (at max value), the baseline fEPSP slope for *Camk2a-Cre;Rng105*^f/f^ mice was set at 45.1% of that for *Rng105*^f/f^ mice in revised Figure 4. Accordingly, we used dual scales to indicate the extent of LTP in *Rng105*^f/f^ mice and *Camk2a-Cre;Rng105*^f/f^ mice (the scale for *Camk2a-Cre;Rng105*^f/f^ mice was 45.1% of that for *Rng105*^f/f^ mice). We have added these descriptions in Figure 4 legend. Because this reviewer's comment is closely related to the next comment, please also refer to our reply to the next comment.

There is no description provided here of the input strength (% of max fEPSP) used for tetanic brain stimulation (TBS). Typically, the stimulus is delivered at 30-50% of the maximum fEPSP for each genotype, it is unclear what was used here. The graph in Figure 4 shows comparable LTP induction post-TBS between WT and KO. There is some text in the last paragraph of the subsection “RNG105 cKO reduces synaptic responses to stimulation” that seems to address the confound here but, it is unclear how the data was processed to arrive at the graph seen in 4F. Does the Y-axis label "% of baseline" refer to the percent of genotype baseline? If so, then this is not ideal because the input/output responses from KO slices is reduced. The convention is to show% of baseline control, which in this case is the WT littermate. This should be clarified because it is very likely that the KO mice do exhibit LTP abnormalities that are masked by this non-standard analysis. Theta burst is a very strong LTM-inducing protocol so these experiments could be masking differences that may be observed with weaker LTP-inducing protocols. If an alternative conclusion is reached, then it could resolve the somewhat confusing dichotomy between observations on late LTP (normal) and "long-term memory," (defective) in RNG105 mutants.

We are grateful for this comment. The input strength was 50% maximum of fEPSP, and we have added the sentence "For LTP recording and theta-burst stimulation, 50% maximum stimulus intensity was used" in Materials and methods. In the initial manuscript, "% of baseline" referred to the percent of each genotype baseline. We believe this is a standard method to compare the extent of synaptic plasticity of genotypes, because of measuring the change in the synaptic efficiency after theta-burst stimulation from the baseline of 50% stimulation of functional synapses in each genotype. However, we agree with the reviewers' comments that it is ideal to rescale the Y-axis for *Camk2a-Cre;Rng105*^f/f^ mice in order to show the difference in synaptic transmission between the genotypes. As described above, the baseline fEPSP slope for *Camk2a-Cre;Rng105*^f/f^ mice was set at 45.1% of that for *Rng105*^f/f^ mice in revised Figure 4. In order to indicate the extent of LTP in each genotype, we used dual scales for *Rng105*^f/f^ mice (left scale) and for *Camk2a-Cre;Rng105*^f/f^ mice (right scale). We consider this type of LTP abnormality, i.e., reduced amplitude and slope of EPSP with normal LTP efficiency and normal late-phase LTP, may be a cause of long-term memory deficits, as also reported in other mice (Discussion, sixth paragraph). Please also refer to our answer to the comment 2 in which we proposed a hypothesis of how this type of LTP abnormality is associated with translation deficiency, spine volume reduction, and impaired long-term memory formation.

4) There are some caveats to the interpretation of behavioral and memory tests that should be addressed.a) The finding that motor learning is normal in RNG105 mutants is consistent with the finding that conditional excision of exon-5 and 6 is less efficient in the cerebellum. But it is equally consistent with the idea that RNG105 is simply not required for this form of learning and memory. This should be acknowledged.

Thank you for this suggestion. We have added the sentence "as well as a notion that RNG105 may be simply not required for this cerebellum-dependent form of learning and memory."

b) The finding that RNG105 mutants show defects in habituation to their environments is interesting. However, the data in Figure 5 should be analyzed by RM-ANOVA not, single trial ANOVA. Could the results be explained by greater activity (increased distance moved) in the cKO rather than a habituation defect? Habituation is better assayed using other tests (EPM, Object Location, Object recognition) and any conclusion on habituation should be strengthened by additional experimentation or moderated by acknowledging caveats.

We are grateful for this suggestion. We have analyzed the data in Figure 5 using two-way repeated measures ANOVA. The ANOVA analysis indicated that the activity of *Camk2a-Cre;Rng105*^f/f^ mice was greater than that of the other genotypes, but this was because of greater activity of *Camk2a-Cre;Rng105*^f/f^ mice in the trial 3 (p = 8.17E-4, one-way ANOVA, cf. Statistical Reporting Table); there was no significant difference in the activity among the genotypes in trials 1 or 2 as judged by one-way ANOVA (Trial 1, p = 0.109; Trial 2, p = 0.0957, cf. Statistical Reporting Table). Taking into account of the statistical significance in the interaction between genotypes and trials, the relative greater activity of *Camk2a-Cre;Rng105*^f/f^ mice than the other genotypes in the trial 3 was not because of their intrinsic activity, but because repeated trials influenced differently on the activity between *Camk2a-Cre;Rng105*^f/f^ mice and the other genotype mice. We have revised a sentence as follows: "In open-field test, there were no differences in exploratory horizontal locomotion among *Rng105*^f/f^, RNG105 hetero-deletion (*Camk2a-Cre;Rng105*^f/+^) and *Camk2a-Cre;Rng105*^f/f^ mice *in the initial trial* (Figure 5)".

In addition, in a light/dark transition test (newly added Figure 5—figure supplement 1), *Camk2a-Cre;Rng105*^f/f^ mice traveled comparable distance to the other genotypes in the chambers, which supported that intrinsic activity of *Camk2a-Cre;Rng105*^f/f^ mice was not greater than that of the other genotypes. According to the suggestion, we further conducted a novel object recognition test. *Rng105*^f/f^ mice showed a high preference for a novel object over a familiar one, whereas *Camk2a-Cre;Rng105*^f/f^ mice did not show such a preference (Figure 5—figure supplement 1), which supported the notion that habituation was impaired in *Camk2a-Cre;Rng105*^f/f^ mice. Taken together, we have added descriptions as follows:

"Habituation is a complex behavior, which is impaired by several factors such as memory deficits, incomplete initial exploration of the entire area because of high level of anxiety, and low level of locomotor activity (Bolivar, 2009). […] These results supported the notion that habituation, a form of learning and memory, was impaired in *Camk2a-Cre;Rng105*^f/f^ mice".

If the authors consider it reasonable, then the Discussion should consider whether increased stress and anxiety could inhibit habituation particularly given that mutations in Caprin 1 are associated with autism.

Thank you for this comment. RNG105 heterozygous mice showed ASD-like behavior, but did not show increased anxiety-like behavior in the light/dark transition test or the elevated plus maze test (Ohashi et al., 2016). *Camk2a-Cre;Rng105*^f/f^ mice have also been subjected to the light/dark transition test (newly added Figure 5—figure supplement 1), which revealed that *Camk2a-Cre;Rng105*^f/f^ mice did not show increased anxiety-like behavior either. We have added new descriptions in the Results section as follows: "Habituation is a complex behavior, which is impaired by several factors such as memory deficits, incomplete initial exploration of the entire area because of high level of anxiety, and low level of locomotor activity (Bolivar, 2009). However, the latter two factors were not likely reasons for the habituation defect in *Camk2a-Cre;Rng105*^f/f^ mice, because the initial exploratory activity was normal in the open field test and anxiety-like behavior in the light/dark transition test was also normal in *Camk2a-Cre;Rng105*^f/f^ mice (Figure 5; Figure 5−figure supplement 1A)".

c) The inability of RNG105 mutants to find the hidden platform in the Morris Maze experiment is confusing. The performance in Figure 6 suggests the mice are incapable of learning the task, making the probe test somewhat pointless (floor effect). Was outlier analysis performed on these data sets? The variance in the KO mice is quite high.

We thank you for this comment. Although escape latency of *Camk2a-Cre;Rng105*^f/f^ mice did not shorten at all (Figure 6), it is not likely to mean that the mice did not learn the task, because density plot of swim path showed weak preference of *Camk2a-Cre;Rng105*^f/f^ mice for the target quadrant (Figure 6). According to the reviewer's suggestion, we have conducted outlier analysis of *Camk2a-Cre;Rng105*^f/f^ mice in each quadrant. Two mice showing maximum values in L and O quadrants (Figure 6) were identified as outliers (p < 0.01 using Smirnov-Grubbs test). These mice remained in the same spots in the quadrants swimming against the wall. One-way repeated measures ANOVA for *Camk2a-Cre;Rng105*^f/f^ mice after elimination of these two mice from the data detected significance in the quadrant effect (F[3,45] = 3.234, p = 0.0309). We have not eliminated the outliers in the revised Figure 6−D, but described these statistical results in the text as follows: "These severe phenotypes raised a concern that *Camk2a-Cre;Rng105*^f/f^ mice might be incapable of learning the task. However, if outliers were eliminated (two *Camk2a-Cre;Rng105*^f/f^ mice showing maximum values in L and O quadrants [Figure 6], p < 0.01 using Smirnov-Grubbs test), statistical significance was detected in the quadrant effect in *Camk2a-Cre;Rng105*^f/f^ mice (one-way repeated measures ANOVA, F[3,45] = 3.234, p = 0.0309). Consistently, the density plot of swim path showed weak preference of *Camk2a-Cre;Rng105*^f/f^ mice for the target quadrant (Figure 6), suggesting *Camk2a-Cre;Rng105*^f/f^ mice were able to learn the task". Because of the existence of weak preference of *Camk2a-Cre;Rng105*^f/f^ mice for the target quadrant after the elimination of the outliers, we have changed the description "RNG105 cKO mice showed no biased preference for either quadrant" to "*Camk2a-Cre;Rng105*^f/f^ mice markedly reduced the time in the target quadrant compared to the other genotypes".

5) The analysis of mRNAs and pathways using GO approaches and subsequent selection of pathways to comment on is typical and not unreasonable, but the discussion and analyses should be expanded to clearly acknowledge alternative unknown and possibly indirect mechanisms by which RNG105 mutations could result in phenotypes that are described.

We thank you for this suggestion. We have added descriptions about analyses on other major GO categories in the Results section as follows: "There are various Arf GAPs and GEFs possessing or not possessing a membrane-associated pleckstrin-homology (PH) domain, among which the Arf GAPs and GEFs identified here were mostly the PH domain-possessing types and classified in "pleckstrin homology" category with other PH domain-containing proteins (Figure 8; Supplementary file 4)"; "Fc γ receptor-mediated phagocytosis", "leukocyte activation", and "chemotaxis" categories also had high-fold enrichment scores, which contained several overlapping proteins such as PI3 kinase pathway proteins and Rac pathway proteins involved in actin regulation, and extracellular membrane proteins (Figure 8; Supplementary file 4)". Because the PI3 kinase pathway was not mentioned in the initial manuscript, we have mentioned it in the Discussion section as follows: "Furthermore, the identified mRNAs included those encoding regulators of Ras, Rho, and the PI3kinase andRacpathway proteins, involved in actin reorganization and spine formation (Nishiyama and Yasuda, 2015; Sala and Segal, 2014)".

Furthermore, we have added discussion about mRNAs for ribosomal proteins as follows: "mRNAs for ribosomal subunit proteins were the major dendritic mRNAs and also reported in the previous studies (Cajigas et al., 2012; Ainsley et al., 2014). However, whether locally translated ribosomal proteins are involved in ribosome biogenesis or in other biological processes, which could be associated with RNG105-defcient phenotypes, remains elusive".

6) There appears to be a section missing in the Results section. It appears that Figure 9 is not referred to in the text? If so, then this figure should be removed or the text for it should be added.

Figure 9 was referred in the initial manuscript. But this was unremarkable as this reviewer pointed out. We have added the text for it: "D-mRNAs included in these categories were plotted on MA plots, which indicated that the dendritic accumulation of these mRNAs was comparable to that of well-known dendritic mRNAs (Figure 9, cf. Figure 7)". In addition, Figure 9 has been referred to in the second paragraph of the subsection “Classification of mRNAs whose localization and expression are changed by RNG105 deficiency”.

7) The premise behind Figure 10 is interesting but the approach used is highly variable and subject to many experimental factors (i.e. staining robustness) that can make this type of analysis challenging especially across multiple cultures. One would be much more confident of this result if it was supported by a more quantitative approach such as biotin surface labeling and immunoblot measurement of labeled proteins. Normalization is much more reliable with this approach. Additionally, the Materials and methods section should be updated to include more information about how ROIs are used and sized, whether 'n' refers to independent cultures or cells in a single culture etc. This is the only mechanistic experiment in the study and the results have important implications for RNG105 function and should ideally be validated using another approach. Finally, there are four groups here and student's t-test is not the right test for this experiment, the variability is very high in the samples and overall effects modest, it would be important to see if significant findings hold up to proper posthoc testing.

We are grateful for this comment. The Materials and methods section, including the information about the selection of ROIs, has been updated as follows: "To count the number of GluR1 and GluR2 puncta, the images were converted into binary images by selecting dendritic regions and using the MaxEntropy threshold algorithm in ImageJ software. […] Fluorescence intensity of GluR1 and GluR2 puncta immunostained before permeabilization was normalized by that immunostained after permeabilization". 'n' refers to the number of neurons. Considering the variability across multiple cultures, we have increased the number of experiments and neurons analyzed in the revised manuscript. The results were essentially the same as before. Accordingly, Figure 10 legend for 'n' has been updated as follows: "In C, n = 31 (*Rng105*^+/+^, −), 35 (*Rng105*^+/+^, +), 34 (*Rng105*^−/−^, −), and 33 (*Rng105*^−/−^, +) neurons from 4 experiments. In D, n = 39 (*Rng105*^+/+^, −), 40 (*Rng105*^+/+^, +), 39 (*Rng105*^−/−^, −), and 38 (*Rng105*^−/−^, +) neurons from 4 experiments."

The data in Figure 10 have been analyzed using two-way ANOVA. In addition, the graph type has been changed to the mean ± s.e.m. Post-hoc t-test indicated that the surface expression of GluR1 and GluR2 was increased in wild-type neurons, but not in RNG105 knockout neurons, after activity deprivation.

According to the reviewer's suggestion, we further conducted biotin surface labeling experiments followed by immunoblot measurement for GluR1 and GluR2 (Figure 10—figure supplement 1). The results were essentially consistent with the results of the immunofluorescence imaging of GluR1 and GluR2, which strongly supported the notion that homeostatic scaling of AMPARs was impaired in RNG105-deficient neurons. Newly added text is as follows:

"We further conducted biotin labeling of cell surface proteins of cultured neurons followed by immunoblot measurement for GluR1 and GluR2 (Figure 10—figure supplement 1). […] These results were consistent with the results of the immunofluorescence imaging of GluR1 and GluR2".

8) The paper ends with the observation that "Thus, this study provides… new knowledge on a physiological function of RNA granules." This is not true at all – as the effect of caprin mutations on granules is never studied, only the function of the Caprin1 protein is analyzed, which could be distinct from the function of an RNA granule.

Thank you for this comment. We agree that this sentence does not correctly describe what we found. We have deleted this sentence.

9) The authors correctly identify a major departure from their findings from similar studies that also examined RNA constituents but did not find major impairments in LTM. However, the authors do not attempt to speculate on why this might be. This is especially important because some of these previous studies examine RNA granule components such as Ataxin-2 that have actually been shown to be required to RNA granule formation. Some discussion to this point would be helpful. Also, there are several papers analysing functions of other granule proteins (e.g. Gld-2, Atx2, Staufen, Pumilio, DDX6/Me31B) on long-term memory that are not cited or discussed, in particular from Drosophila work.

We are grateful for this comment. We have mentioned knockout mice for GLD-2 and Ataxin-2 in the revised manuscript (Discussion, second paragraph). Although a previous report demonstrated that knockdown of Ataxin-2 in cultured cells markedly reduced the number and size of RNA (stress) granules (Nonhoff et al., Mol. Biol. Cell, 2006), a recent report demonstrated that knockdown of Ataxin-2 in cultured cells only delayed the maturation of stress granules (Becker et al., Nature, 2017). Therefore, we have not discussed about the requirement of Ataxin-2 and RNA granule formation in the revised manuscript. We have cited and discussed the effects of mutations of RNA granule components in mice and *Drosophila* as follows:

"The different phenotypes between RNG105 conditional deletion mice and the other KO mice of RNA granule components may be attributed to the existence and non-existence of alternative factors. […] Thus, RNG105 deficiency may have large impact on the function of RNA granules and therefore the formation of long-term memory".

10) FMRP mutants in Drosophila have been reported to have specific defects in long-term memory. Could the different observation in mammals be due to the existence of alternative isoforms? How many isoforms of RNG105 and G3BP exist in rodents and how much could or do their functions overlap in vivo? This issue should be broadly considered in the Discussion as moderating caveat to the mutant analysis described.

We are grateful for this comment. As the reviewer suggested, we consider the different effects of RNA granule mutants on long-term memory between mice and *Drosophila* may be attributed to the existence and non-existence of alternative isoforms and/or cooperative activity of the isoforms. We have added discussion about the effects of mutants of RNA granule components possessing and not possessing alternative factors on long-term memory in mice and *Drosophila* as described above in the answer to the comment 9.

Mice have two RNG/caprin isoforms (RNG105/caprin1 and RNG140/caprin2) and two G3BP isoforms (G3BP1 and G3BP2). G3BP1 and G3BP2 appear to have redundant functions because stress granule formation in double knockdown cells was much more impaired than in single knockdown cells (Matsuki et al., Genes Cells, 2013). In contrast, RNG105 and RNG140 may not functionally overlap, because they are localized to different kinds of RNA granules, and knockdown phenotypes of RNG105 and RNG140 in cultured neurons were not compensated by each other (Shiina and Tokunaga, J. Biol. Chem., 2010). We consider this difference in the existence of alternative factors may underlie the difference in long-term memory formation. Again, please refer to our reply to the comment 9.